# GraphPFN: A Prior-Data Fitted Graph Foundation Model

**Dmitry Eremeev** [1] [2]  **Oleg Platonov** [1] [2]  **Gleb Bazhenov** [1] [2]  **Artem Babenko** [2]  **Liudmila Prokhorenkova** [2]

## Abstract

Graph foundation models face several fundamental challenges including transferability across diverse domains and data scarcity, which calls into question the very feasibility of creating such models. However, despite similar challenges, the tabular domain has recently witnessed the emergence of the first successful foundation models such as TabPFN. These models are based on the prior-data fitted networks (PFN) framework, in which models are pretrained on carefully designed synthetic datasets to make predictions in an in-context learning setting. Recently, G2T-FM, a framework that converts graph node-level tasks into tabular tasks, has made the first step towards adopting PFNs for graphs, yet it is limited to hand-crafted features and was never pretrained on graph data. In this work, we make the next step by proposing GraphPFN, a PFN-based model designed and pretrained specifically for graph node-level tasks. Following the PFN framework, we first design a prior distribution of synthetic attributed graphs by using a novel combination of multi-level stochastic block models and a preferential attachment process for structure generation and graph-aware structured causal models for attribute generation. Then, we augment the tabular foundation model LimiX with attention-based graph neighborhood aggregation layers and train it on millions of synthetic graphs sampled from our prior. On diverse real-world graph datasets with node-level tasks, GraphPFN achieves state-of-the-art results in both in-context learning and finetuning regimes, outperforming G2T-FM, prior GFMs, and task-specific GNNs trained from scratch. More broadly, GraphPFN shows the potential of PFN-based models for building graph foundation models. Our code is available at https://github.com/yandex-research/graphpfn.

[1]HSE University [2]Yandex Research. Correspondence to: Dmitry Eremeev <eremeev-d@yandex-team.ru>.

*Proceedings of the 43rd International Conference on Machine Learning*, Seoul, South Korea. PMLR 306, 2026. Copyright 2026 by the author(s).

## 1. Introduction

In recent years, foundation models have significantly advanced the state of the art in domains such as natural language processing and computer vision. These models can learn from large unannotated datasets and generalize across a wide range of downstream tasks. Notable early examples include BERT (Devlin et al., 2019) and GPT-3 (Brown et al., 2020) in natural language processing, as well as ViT (Dosovitskiy et al., 2020) and CLIP (Radford et al., 2021) in computer vision. These models often use self-supervised or unsupervised pretraining to learn rich, transferable representations. This approach has changed how models are built and used, removing the need for task-specific models and reducing the dependence on large labeled datasets for every single task. Inspired by these successes, there is growing interest now in extending the foundation models methodology to other modalities, including graphs.

However, developing graph foundation models (GFMs) is much more challenging. Unlike text and images, graph data does not constitute *a single domain*. Instead, graphs are used to represent data from *different domains*, e.g., social networks (both virtual and real-world), information networks, transportation networks, co-purchasing networks, various physical, biological, or engineering systems, or even networks of abstract concepts. As a consequence, both the structure of a graph and its attributes (features and labels) may vary significantly across graph datasets and tasks. Moreover, the amount and diversity of the available graph data are significantly lower compared to those in computer vision and natural language processing. Taken together, these challenges call into question the very feasibility of graph foundation models.

Despite facing similar challenges, tabular machine learning has recently witnessed the emergence of the first successful tabular foundation models (TFMs) such as TabPFNv2 (Hollmann et al., 2025) and LimiX (Zhang et al., 2025b). Many of these models are based on the framework of prior-data fitted networks (PFNs, Müller et al., 2022; Hollmann et al., 2023). PFNs are pretrained on diverse synthetic datasets drawn from a *prior* to approximate Bayesian inference and can make predictions via in-context learning. Strong performance in the tabular domain suggests that PFNs offer a promising path towards building foundation models for

both tabular and graph domains.

The recent works G2T-FM (Eremeev et al., 2025), TAG (Hayler et al., 2025), and TabPFN-GN (Choi et al., 2025) have made the first step towards adopting PFNs for graph tasks by utilizing foundation models for tabular data to create graph foundation models for node-level tasks. For this, they augment node features with graph-based information such as neighborhood-aggregated features or Laplacian positional encodings. This allows for transforming a graph node-level prediction problem into a tabular prediction problem and applying an existing tabular foundation model to this task. The resulting approach shows strong performance, but such models still depend heavily on hand-crafted features and lack large-scale pretraining on diverse graph data. As a result, they are limited in their ability to capture complex graph patterns.

In this work, we make the next step by proposing GraphPFN, a PFN-based model designed and pretrained specifically for graph node-level tasks. Following the PFN framework, we pretrain GraphPFN on synthetic datasets drawn from a carefully designed graph prior. For generating graph structures, we propose an approach that combines multiple stochastic block models and augments them with a preferential-attachment process. We then generate graph-structure-dependent node attributes for our graphs by augmenting tabular structured causal models (SCMs, Hollmann et al., 2023; Qu et al., 2025) typical for tabular PFNs with message-passing mechanisms at random SCM nodes. This method allows us to efficiently generate millions of realistic and diverse synthetic graph datasets. Then, we initialize GraphPFN from a PFN-based tabular foundation model LimiX (Zhang et al., 2025b) and add an attention-based message-passing layer to each block of LimiX. This allows the model to learn complex graph-specific patterns while retaining its ability to handle diverse features and labels inherited from LimiX.

Our experiments show that on diverse real-world graph node-level prediction datasets, GraphPFN achieves state-of-the-art in-context learning performance, outperforming, on average, the best current models — well-tuned traditional GNNs (Kipf & Welling, 2017; Hamilton et al., 2017; Veličković et al., 2018; Shi et al., 2021) with improved architectures (Platonov et al., 2023b) and recent TFM-based GFMs, G2T-FM (Eremeev et al., 2025) and TAG (Hayler et al., 2025). Finetuning further boosts performance, allowing GraphPFN to achieve the best results on 12 out of 13 datasets, often with a substantial improvement over the second-best model.

Overall, our main contributions are as follows:

- We propose GraphPFN, the first publicly available PFN-based model designed and pretrained specifically for graph node-level tasks.

- We introduce a novel graph prior for the efficient generation of realistic synthetic attributed graphs.

- We demonstrate that GraphPFN achieves state-of-the-art results in both in-context learning and finetuning regimes.

More generally, our study shows that pretraining graph-aware PFNs on synthetic graph datasets from a well-designed graph prior is a promising direction for building powerful and generalizable graph foundation models.

## 2. Related Work

### 2.1. Prior-Data Fitted Networks

Prior-data fitted networks (PFNs) were first introduced by Müller et al. (2022). The main idea behind PFNs is to train models that can make predictions on previously unseen datasets in a single forward pass. These models leverage in-context learning (ICL): rather than updating model parameters for each new dataset, they use the context provided at inference time to adapt their predictions without additional training.

In the PFN framework, the input to the model consists of two parts: a set of training samples with their labels, called the *context*, and a set of test samples without labels, called the *query*. During a single forward pass, the model uses the context to make predictions for the query samples, thus performing ICL. In practice, PFNs are commonly implemented as Transformers with a specific attention mask (Hollmann et al., 2023; 2025): context (training) samples attend to all other context samples, while query (test) samples are only allowed to attend to context samples and not to each other. This structure ensures that predictions for each query are based solely on the training data.

PFNs are trained via pretraining on a large collection of synthetic datasets. To achieve this, one specifies a *prior* over supervised datasets, and the model is trained to perform ICL as described above, predicting labels for query samples given context samples. As shown by Müller et al. (2022), this procedure trains the network to approximate the posterior predictive distribution under the chosen prior, which provides the main theoretical motivation for the approach.

### 2.2. Tabular Foundation Models

In their pioneering work TabPFN (Hollmann et al., 2023) and its successor TabPFNv2 (Hollmann et al., 2025), the authors proposed to utilize the framework of prior-data fitted networks to create tabular foundation models and showed that such models can achieve strong results, competitive with other approaches (Erickson et al., 2025). Nowadays, TFMs have become an active research area, with several new methods (mostly based on the PFN framework) re-

leased recently (Qu et al., 2025; 2026; Zhang et al., 2025a;b; Grinsztajn et al., 2025; 2026). Some methods focus on scalability (Qu et al., 2025) or faster inference (Mueller et al., 2025), while others emphasize training on real-world datasets rather than synthetic tasks (Ma et al., 2025). Together, these works broaden the design space of tabular foundation models by trading off data sources, computational efficiency, and scalability.

## 2.3. Graph Foundation Models

Similar to foundation models for tabular data, graph foundation models face the challenge of handling datasets from diverse domains. A particularly difficult, yet crucial, aspect is managing the wide variety of node attributes (features and labels) present in different graphs. Early GFMs did not fully address this issue. They often relied on dimensionality reduction techniques such as principal component analysis or singular value decomposition (Xia & Huang, 2024; Zhao et al., 2024; Wang et al., 2025; Yu et al., 2025), or restricted their focus to graphs where node attributes are all textual (Liu et al., 2024; Wang et al., 2024; He et al., 2025).

More recent works, such as G2T-FM (Eremeev et al., 2025), TAG (Hayler et al., 2025), and TabPFN-GN (Choi et al., 2025), have explored leveraging tabular foundation models to better address feature diversity in graph datasets. These approaches incorporate hand-crafted features, for example, neighborhood-aggregated features or Laplacian positional encodings, to effectively convert graph information into tabular features. Empirical results show that these methods achieve strong results, always significantly outperforming prior GFMs (such as those introduced in Xia et al. (2024); Xia & Huang (2024); Zhao et al. (2024); Yu et al. (2025); Wang et al. (2025); Finkelshtein et al. (2025)) and frequently outperforming well-tuned classic GNNs (Kipf & Welling, 2017; Hamilton et al., 2017; Veličković et al., 2018; Shi et al., 2021) which are trained from scratch on each dataset and use improved architectures from Platonov et al. (2023b), thus supporting the utility of employing tabular foundation model approaches as a basis for learning on graph data.

## 3. GraphPFN

GraphPFN is a foundation model designed for in-context learning or finetuning on graph-structured data. Inspired by recent advances in prior-data fitted networks (PFNs) for tabular data (Hollmann et al., 2025; Zhang et al., 2025b), GraphPFN extends these ideas to graphs by augmenting a tabular foundation model with attention-based message-passing adapters. This design allows GraphPFN to reuse strong feature modeling from tabular pretraining while capturing complex graph-specific patterns.

### 3.1. Architecture

Our model architecture extends a PFN-based tabular foundation model LimiX (Zhang et al., 2025b) by adding attention-based message-passing layers to each of its transformer blocks. This strategy is inspired by recent findings that tabular foundation models can already learn patterns relevant for various graph tasks (Eremeev et al., 2025; Hayler et al., 2025; Choi et al., 2025). By initializing our model with a pretrained tabular foundation model instead of training from scratch, we leverage these learned representations, which significantly reduces computational costs while still achieving strong results.

Below, we first summarize the common architecture of current PFN-based tabular foundation models (Hollmann et al., 2025; Zhang et al., 2025b), considering LimiX (Zhang et al., 2025b) as a specific example, to clarify how GraphPFN represents samples (nodes) and features, and how representations and attention flow in the base model. We then describe how the graph adapters modify these flows to leverage the graph topology.

**LimiX**  LimiX (Zhang et al., 2025b) is a transformer-style foundation model for tabular data that, following TabPFNv2 (Hollmann et al., 2025), departs from the common design of representing each sample with a single fixed-length embedding. Instead, it uses a multi-token representation: for every sample, each feature contributes one token,[1] yielding a token grid with one axis for features and one for samples. This design naturally handles a variable number of heterogeneous features in different datasets without changing and retraining the model.

A LimiX transformer block contains three attention layers, each followed by an element-wise feed-forward network (FFN). Two layers are feature-level multi-head attention (MHA) modules that operate within a sample, allowing all feature tokens of the same sample to attend to each other. The third is a sample-level MHA that operates within a feature across samples, allowing tokens corresponding to the same feature to exchange information across the dataset. The two feature-level MHAs enable rich interactions among features within each sample, while the sample-level MHA allows sample interaction by transferring information across samples for the same feature.

Attention masking at the sample level follows the standard PFN protocol: training (context) samples attend to all other training samples, and test (query) samples attend only to training samples. Thus, information can flow from train to test but not from test to train or between test samples.

---

[1]More precisely, for better efficiency, features are grouped into pairs, and each feature pair contributes one token, but we omit this detail for clarity.

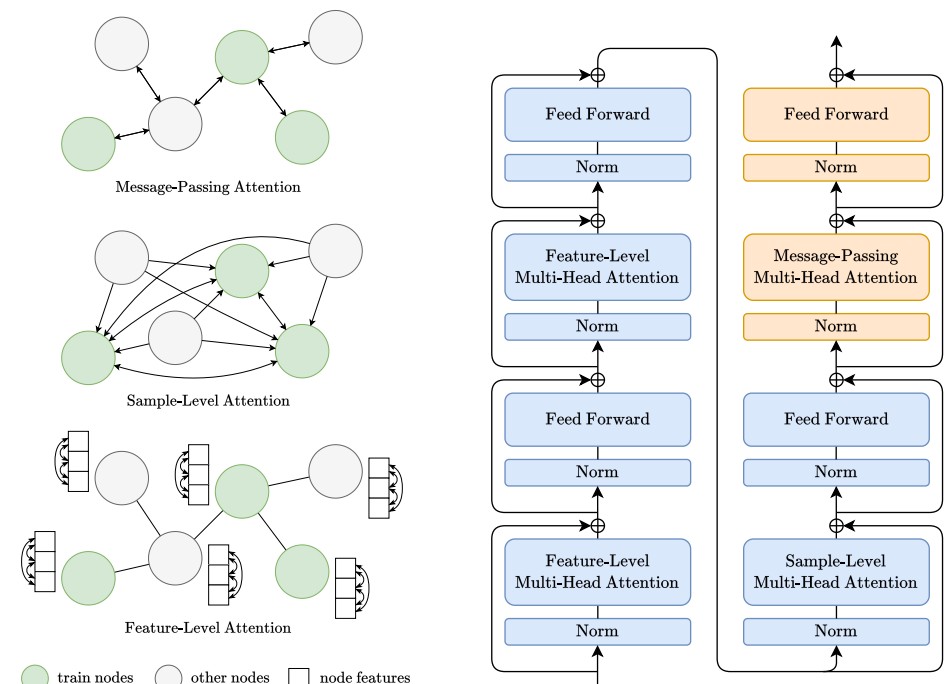

*(a)* Different attention mechanisms in GraphPFN.   *(b)* Base neural block in GraphPFN architecture.

*Figure 1.* An illustration of GraphPFN architecture.

Feature-level attention within a sample is unmasked.

**Graph adapters**   To inject graph structure information without disrupting LimiX's tokenization, we add a message-passing adapter that implements scaled dot product attention between neighboring nodes at the end of every LimiX transformer block (see Figure 1 for an illustration). Note that we use the scaled dot product attention that is identical to the original Transformer attention (Vaswani et al., 2017) except for being restricted to 1-hop graph neighborhoods. Intuitively, our message-passing adapter performs a second, graph-structure-aware round of sample-level attention: tokens may exchange information only along the observed edges. In contrast to the global sample-level attention of LimiX (which is masked by the PFN protocol), the graph adapter is masked by the adjacency and therefore routes information locally, from each node to its neighbors. Because we keep the per-feature token representation intact, the adapter runs over the sample axis for each feature token independently, using the same graph mask across features.

This design complements the global PFN-style attention by adding a local graph-structure-based message-passing channel that is common in graph machine learning. We process the entire graph jointly and the graph adapter allows bidirectional exchange between labeled and unlabeled nodes along edges. Similar to the classic GNNs, these message-passing layers allow the model to capture complex graph dependen-

cies that cannot be captured by hand-crafted features.

Each adapter is implemented as a sparse, multi-head attention module with the adjacency as its mask, where two nodes attend to each other if and only if an edge connects them. Similar to other attention modules, it is followed by a feed-forward network independently applied to each token. Both the FFN and the message-passing layer are wrapped with residual connections (He et al., 2016) and layer normalization (Ba et al., 2016), mirroring the structure of the LimiX blocks for stable optimization.

### 3.2. Pretraining

GraphPFN was pretrained under the PFN framework (Müller et al., 2022; Hollmann et al., 2023; 2025) by continuing training from the LimiX checkpoint. In total, we used 1,600,000 synthetic datasets generated according to the prior described below. Pretraining was conducted for 10,000 optimizer steps and ran for approximately 36 hours on 8 NVIDIA A100 80GB GPUs. Each GPU processed one synthetic dataset per step with 20 gradient accumulation steps, resulting in 160 datasets per optimizer step.

We optimized the model using AdamW (Kingma & Ba, 2015; Loshchilov & Hutter, 2019) with weight decay weight of 0.1. We used a cosine annealing learning rate schedule (Loshchilov & Hutter, 2017) with linear warmup over the first 10% of training steps and a base learning rate of

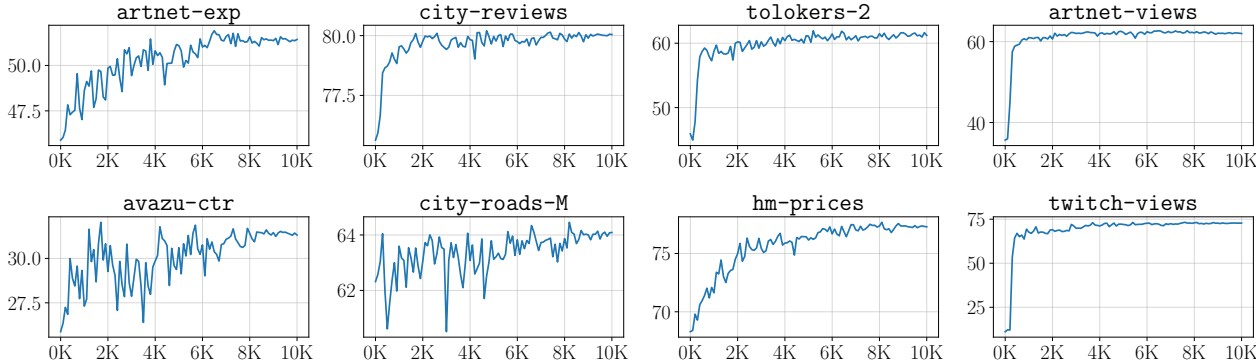

*Figure 2.* In-context learning performance of intermediate checkpoints of GraphPFN. The $x$-axis represents the number of steps, and the $y$-axis represents the metric on the test set. Evaluation is performed with random features, but without ensembling.

$10^{-3}$. To improve training stability and preserve the feature modeling capabilities of LimiX, we froze all model layers except the graph adapters. Only these components were updated during pretraining. To further stabilize training, we applied an exponential moving average (EMA) to the adapter weights throughout pretraining, with decay of $0.98$.

**Objective** We optimize a joint objective that combines the PFN supervised loss with the masked graph modeling (MGM) loss (Li et al., 2023). For the supervised PFN term, we sample a random set of context nodes for each dataset, compute predictions for all other nodes, and minimize the supervised loss on them. We use the cross-entropy loss for classification tasks and the mean squared error loss for regression tasks.

To encourage more graph-aware representations, inspired by the success of GNN self-supervised pretraining, we add the masked graph modeling term from Li et al. (2023). Specifically, we randomly sample a fraction $p = 0.1$ of edges as positive samples, remove these edges from the input graph, and uniformly sample an equal number of unconnected node pairs as negative examples. We then train the model with the cross-entropy loss to distinguish between positive and negative edges. For this, we apply an additional MLP head to the elementwise multiplication of the target embeddings from the last layer for the source and destination nodes of each sampled edge. The total loss is the sum of the supervised loss and the MGM loss, with a weight of $0.1$ applied to the MGM term.

**Learning curves** To monitor the progress of pretraining, we evaluated the in-context learning performance of GraphPFN on several datasets from the GraphLand benchmark (Bazhenov et al., 2025) (see Section 5.1 for the evaluation procedure) every 100 steps. We emphasize that these evaluations were used only for a post-hoc analysis of training progress: test set performance was not used for early

stopping or any other form of model selection.

Figure 2 shows that the first 1–2K steps enable GraphPFN to quickly reach reasonable performance. In particular, on datasets such as `artnet-views`, `tolokers-2`, and `twitch-views`, where the performance gap between the graph-agnostic LimiX and GraphPFN is largest, most of the gap is closed within the first 1–2K gradient steps. Nevertheless, further pretraining continues to improve performance on average and is essential for achieving strong final performance of GraphPFN.

## 4. Graph Prior

As discussed above, GraphPFN is based on the prior-data fitted networks (PFNs) framework (Müller et al., 2022). In this approach, the model is pretrained on a large number of synthetic graph datasets sampled from a chosen prior distribution. Since the pretrained model aims to approximate the posterior predictive distribution, it is crucial to design a diverse, high-quality, and realistic prior. In this section, we describe the prior used for pretraining GraphPFN. First, we explain our method for generating realistic graph structures. Then, we describe how we use these graphs to generate node attributes and targets.

### 4.1. Structure Generation

Our main aim is to generate graph structures similar to real-world graphs. After examining graphs from a range of graph machine learning datasets, we find that most of them exhibit strong clustering (community) structure, which in general is a common feature of real-world graphs (Girvan & Newman, 2002). Thus, as the basis for our graph generation process, we use the degree-corrected stochastic block model (SBM) (Karrer & Newman, 2011), which can generate graphs with community structure. However, we find that graphs generated from the degree-corrected SBM

exhibit clusters that appear overly clean and well-defined, visually resembling well-separated spheres. In contrast, clusters in real-world graphs are often less well-structured, with more complex shapes and frequent overlaps. Thus, to obtain graph structures similar to real-world ones, we design a novel method that combines multiple SBMs. First, we generate several *first-level* graphs from SBMs with different parameters. Then, we generate a *second-level* graph from another SBM, such that this second-level graph has the number of nodes equal to the sum of the numbers of nodes in all first-level graphs. We then randomly assign each node from the first-level graphs to a unique node in the second-level graph, thus essentially constructing a bijection $f$ between first-level and second-level graph nodes. Finally, we combine the two layers into one graph: we add an edge between the node $u$ and $v$ if either this edge exists in the first-level graphs or the edge between $f(u)$ and $f(v)$ exists in the second-level graph. The obtained graph combines multiple graphs generated from different SBMs and exhibits clusters of nodes with complex shapes and overlaps that we aimed to capture.

Further, we observe that most graphs from graph benchmarks exhibit a core-periphery structure, where the core is composed of multiple relatively dense clusters, but there are also many low-degree peripheral nodes. Such a structure is known to be common in real-world networks (Zhang et al., 2015). While our method of combining multiple graphs generated from SBMs produces realistic node clusters, it produces a relatively small number of peripheral nodes. Thus, we augment the approach discussed above with a preferential attachment (PA) process (Price, 1965; 1976; Albert & Barabási, 2002). Specifically, we use the graph obtained thus far as the initialization for the PA process. We sequentially add low-degree nodes and connect them to the previous ones with probabilities proportional to their degrees. In contrast to the standard PA process which fixes the initial degree of a new node, we choose the initial degree of each new node randomly.

Our method has many hyperparameters such as the number and size of blocks for SBMs or their degree sequences. Similar to prior works on PFNs (Hollmann et al., 2023; 2025; Qu et al., 2025; 2026), we define probability distributions for each hyperparameter and sample new hyperparameters for each synthetic graph, which allows us to generate diverse graphs. At the same time, we can easily set bounds for sizes, densities, or maximum degrees of the generated graphs, allowing us to ensure that the graphs fit the desired constraints.

We provide example visualizations of several graphs generated by our process in Appendix B.1. We also analyze how training on simpler random graphs affects the downstream performance (Appendix B.3). Finally, we inspect the align-ment of various characteristics between real and synthetic datasets in Appendix B.2.

### 4.2. Attribute Generation

We generate features and targets for synthetic graphs with a neural structural causal model (SCM) that extends the MLP-based SCM of TabICL from Qu et al. (2025) (which is in turn inspired by Hollmann et al. (2023)). As a starting point, we follow the TabICL protocol: we sample an MLP architecture (number of layers, dimension of hidden layers, activation function) and its weights at random, draw random inputs, propagate them through the network, and then designate a random subset of neurons as observed features and another random neuron as the target, leaving the rest as latent variables. This yields a broad family of causal mechanisms in which features and targets can depend on each other and on latent confounders. We refer to Qu et al. (2025); Hollmann et al. (2023) for further details.

To make attributes also depend on the graph structure, we extend this SCM in two complementary ways. First, we introduce a mixture of MLP and GNN neurons. For each dataset, we sample a mixing probability $p \in \{0.0, 0.1, \ldots, 0.9, 1.0\}$ and a GNN aggregation type from $\{\text{Mean}, \text{Max}, \text{Min}, \text{GCN}, \text{GT}\}$. At every hidden layer, we compute the outputs of both an MLP transformation and a GNN layer. Each neuron is then independently assigned to be MLP-type or GNN-type, with probabilities $1 - p$ and $p$, respectively, and its value is taken from the corresponding transformation. This mechanism controls how strongly the generated variables depend on the graph. Second, we augment the random input features with structural features. Specifically, we consider node degree and PageRank. Each feature is independently added with probability $0.5$.

Together, the mixed MLP/GNN neurons and optional structural features inject graph information into the SCM while remaining close to the tabular prior. When $p = 0$ and structural features are not used, the procedure reduces to the TabICL-style tabular SCM, while a larger value of $p$ and the inclusion of structural features increase the influence of graph structure on both features and targets.

## 5. Experiments

### 5.1. Experimental Setup

**Datasets** In terms of dataset selection, we follow the experimental setup of G2T-FM (Eremeev et al., 2025). We evaluate two collections of datasets: (i) real-world datasets from the recently proposed GraphLand benchmark (Bazhenov et al., 2025); and (ii) some of the classic graph datasets. Together, these datasets cover node classification and regression, come from diverse application domains, include both homophilous and non-homophilous

*Table 1.* Evaluation results on the GraphLand datasets under the `RL` (Random Low) data split. ICL stands for in-context learning, FT stands for finetuning. We report average precision for binary classification tasks and $R^2$ for regression tasks. We also report average rank over classification datasets in the 'AR (cls)' column and over all datasets in the 'AR (all)' column. During the computation of 'AR (all)' we exclude all methods that do not support regression tasks, so that ranks are computed only over rows with no missing entries. For each column, we highlight the first, second, and third best results with color.

| | artnet-exp | city-reviews | tolokers-2 | artnet-views | avazu-ctr | city-roads-M | hm-prices | twitch-views | AR (cls) | AR (all) |
|---|---|---|---|---|---|---|---|---|---|---|
| LightGBM + NFA | 46.13 ± 0.04 | 78.53 ± 0.01 | 56.34 ± 0.06 | 56.10 ± 0.02 | 31.71 ± 0.01 | 61.18 ± 0.03 | 70.84 ± 0.04 | 60.14 ± 0.01 | 7.33 | 6.00 |
| GCN | 44.86 ± 0.36 | 77.81 ± 0.15 | 56.27 ± 0.31 | 56.03 ± 0.25 | 32.00 ± 0.16 | 58.82 ± 0.25 | 68.02 ± 0.42 | 75.51 ± 0.05 | 9.67 | 6.88 |
| GraphSAGE | 45.14 ± 0.36 | 78.17 ± 0.10 | 54.43 ± 0.34 | 49.32 ± 0.91 | 31.44 ± 0.16 | 59.44 ± 0.27 | 70.00 ± 0.74 | 66.29 ± 0.32 | 8.67 | 7.62 |
| GAT | 45.06 ± 0.52 | 77.74 ± 0.21 | 57.41 ± 0.85 | 53.60 ± 0.24 | 32.63 ± 0.17 | 59.86 ± 0.20 | 72.07 ± 1.22 | 72.89 ± 0.27 | 8.67 | 5.88 |
| GT | 46.41 ± 0.71 | 77.34 ± 0.21 | 56.98 ± 0.55 | 53.37 ± 0.46 | 31.11 ± 0.49 | 59.55 ± 0.28 | 69.44 ± 0.94 | 72.13 ± 0.13 | 8.67 | 7.25 |
| OpenGraph (ICL) | 15.16 ± 0.83 | 59.09 ± 0.72 | 40.38 ± 1.13 | – | – | – | – | – | 13.67 | – |
| AnyGraph (ICL) | 12.84 ± 0.93 | 63.71 ± 1.45 | 28.75 ± 3.56 | – | – | – | – | – | 15.67 | – |
| TS-GNN (ICL) | 20.44 ± 1.05 | 43.46 ± 5.17 | 38.54 ± 0.94 | – | – | – | – | – | 13.67 | – |
| SAMGPT (FT) | 16.20 ± 0.93 | 45.91 ± 2.68 | 36.93 ± 1.07 | – | – | – | – | – | 14.33 | – |
| MDGFM (FT) | 16.99 ± 1.75 | 31.24 ± 8.50 | 31.97 ± 1.34 | – | – | – | – | – | 15.00 | – |
| GCOPE (FT) | 14.92 ± 1.56 | 67.16 ± 0.98 | 28.81 ± 1.28 | – | – | – | – | – | 14.67 | – |
| G2T-LimiX (ICL) | 48.52 ± 0.29 | 77.97 ± 0.53 | 61.58 ± 0.32 | 60.96 ± 0.10 | 32.41 ± 0.14 | 64.69 ± 0.05 | 74.97 ± 0.06 | 71.30 ± 0.07 | 4.67 | 4.38 |
| TAG-TabPFNv2 (ICL) | 47.87 ± 0.39 | 77.38 ± 0.29 | 59.33 ± 1.00 | – | – | – | – | – | 7.33 | – |
| TAG-LimiX (ICL) | 50.19 ± 0.46 | 78.87 ± 0.13 | 59.46 ± 1.05 | – | – | – | – | – | 4.00 | – |
| GraphPFN (ICL) | 51.79 ± 0.11 | 80.25 ± 0.05 | 61.29 ± 0.12 | 62.79 ± 0.08 | 31.63 ± 0.06 | 64.85 ± 0.13 | 77.88 ± 0.09 | 73.20 ± 0.08 | 2.33 | 3.12 |
| G2T-LimiX (FT) | 49.96 ± 0.09 | 80.15 ± 0.07 | 60.71 ± 0.57 | 62.11 ± 0.06 | 33.86 ± 0.33 | 65.96 ± 0.11 | 76.30 ± 0.19 | 73.62 ± 0.49 | 3.67 | 2.88 |
| GraphPFN (FT) | 53.49 ± 0.81 | 80.90 ± 0.03 | 62.80 ± 0.39 | 65.35 ± 0.06 | 35.07 ± 0.34 | 67.30 ± 0.21 | 81.06 ± 0.24 | 79.00 ± 0.14 | 1.00 | 1.00 |

graphs,[2] and span a range of densities and other graph structural properties. Table 10 lists the datasets used in our study and summarizes their statistics. For all datasets, we use 10%/10%/80% train/validation/test splits. Due to the current limitations of TFMs, which often natively support classification tasks with at most 10 classes, we restrict our main experiments to small- and medium-scale datasets and exclude classification datasets with more than 10 classes. However, GraphPFN can be applied to datasets with more classes via error-correcting output codes (ECOC) (Hollmann et al., 2025), and we report additional results for this setting in Appendix C.

In our evaluation, we run all experiments 10 times (5 times for non-PFN-based GFMs) and report the mean and standard deviation of the model performance. We report average precision for binary classification tasks, accuracy for multiclass classification tasks, and $R^2$ for regression tasks. For all metrics, higher is better.

**Methods** In addition to the proposed GraphPFN, we evaluate the following methods:

- **LightGBM** (Ke et al., 2017), a strong tabular baseline, augmented with neighborhood feature aggregation (NFA) (Bazhenov et al., 2025) to incorporate information about the graph-feature interaction.

- **Classic GNNs**: GCN (Kipf & Welling, 2017), GraphSAGE (Hamilton et al., 2017), GAT (Veličković et al., 2018), neighborhood-attention Graph Transformer (GT) (Shi et al., 2021). Following Platonov et al. (2023b), we augment these models with residual connections (He

---

[2]A graph is called homophilous if its edges tend to connect nodes with similar labels, see Newman (2003); Platonov et al. (2023a); Mironov & Prokhorenkova (2024) for details.

et al., 2016), layer normalization (Ba et al., 2016), and MLP blocks, which have been shown to substantially improve the performance of classic GNNs (Platonov et al., 2023b; Luo et al., 2024; 2025). We perform extensive hyperparameter tuning for these models.

- **Prior GFMs:** OpenGraph (Xia et al., 2024), AnyGraph (Xia & Huang, 2024), GCOPE (Zhao et al., 2024), SAMGPT (Yu et al., 2025), MDGFM (Wang et al., 2025), and TS-GNN (Finkelshtein et al., 2025), which are all GFMs not based on the PFN framework.

- **Recent PFN-based GFMs:** G2T-FM (Eremeev et al., 2025) and TAG (Hayler et al., 2025). To the best of our knowledge, these are the strongest publicly available graph foundation models for node-level tasks with arbitrary features.

We evaluate GraphPFN in both in-context learning (ICL) and full finetuning (FT) settings. Other GFMs are evaluated in ICL, FT, or both settings based on their official implementations. For the FT setting, we finetune the entire model, following Eremeev et al. (2025); Rubachev et al. (2025).

Following a common practice in tabular PFNs (Hollmann et al., 2025; Qu et al., 2026), we use inference-time ensembling for G2T-FM, TAG, and GraphPFN. For G2T-FM and GraphPFN, we average predictions from 10 forward passes with different random seeds. These seeds affect random feature permutations and other sources of model randomness, such as random positional encodings in TFMs and PEARL inputs in G2T-FM. We ablate the influence of ensembling in Appendix C. Inspired by Abboud et al. (2020); Sato et al. (2021); Kanatsoulis et al. (2025), we additionally augment the input of GraphPFN with 8 random features during each forward pass in all experiments. For further details on the experimental setup, please refer to Appendix D.

*Table 2.* Evaluation results on classic graph datasets under a random 10%/10%/80% train/val/test split. ICL stands for in-context learning, FT stands for fine-tuning. We report average precision for binary classification tasks and accuracy for multiclass classification tasks. We also report average rank (AR) in the last column. For each column, we highlight the first, second, and third best results with color.

| | amazon-ratings | facebook | pubmed | questions | wiki-cs | **AR** |
|---|---|---|---|---|---|---|
| GCN | $41.43 \pm 0.49$ | $91.26 \pm 0.21$ | $85.46 \pm 0.19$ | $15.42 \pm 0.67$ | $81.74 \pm 0.21$ | **8.40** |
| GraphSAGE | $40.07 \pm 0.53$ | $91.12 \pm 0.22$ | $86.04 \pm 0.27$ | $16.55 \pm 0.64$ | $81.50 \pm 0.27$ | **8.40** |
| GAT | $40.67 \pm 0.55$ | $92.61 \pm 0.21$ | $84.81 \pm 0.24$ | $16.75 \pm 0.67$ | $82.25 \pm 0.27$ | **6.80** |
| GT | $41.56 \pm 0.40$ | $91.71 \pm 0.22$ | $84.95 \pm 0.19$ | $14.03 \pm 0.90$ | $82.54 \pm 0.21$ | **7.80** |
| OpenGraph (ICL) | $29.36 \pm 1.24$ | $75.27 \pm 5.05$ | $70.30 \pm 2.67$ | $3.77 \pm 0.65$ | $75.66 \pm 0.39$ | **13.60** |
| AnyGraph (ICL) | $33.49 \pm 3.44$ | $61.17 \pm 8.64$ | $65.31 \pm 6.26$ | $4.27 \pm 0.66$ | $65.17 \pm 2.51$ | **14.20** |
| TS-GNN (ICL) | $43.00 \pm 0.13$ | $77.87 \pm 2.73$ | $64.41 \pm 5.11$ | $5.00 \pm 0.48$ | $46.25 \pm 9.77$ | **13.20** |
| SAMGPT (FT) | $21.20 \pm 1.90$ | $80.61 \pm 0.93$ | $71.93 \pm 1.45$ | $8.24 \pm 0.48$ | $63.82 \pm 2.35$ | **13.00** |
| MDGFM (FT) | $21.37 \pm 1.34$ | $70.30 \pm 3.59$ | $67.54 \pm 1.94$ | $10.55 \pm 0.50$ | $58.71 \pm 3.74$ | **14.00** |
| GCOPE (FT) | $39.90 \pm 0.43$ | $85.08 \pm 0.17$ | $79.35 \pm 0.70$ | $6.59 \pm 0.43$ | $59.13 \pm 1.20$ | **12.20** |
| G2T-LimiX (ICL) | $44.65 \pm 0.18$ | $91.66 \pm 0.05$ | $89.19 \pm 0.18$ | $15.46 \pm 0.93$ | $81.09 \pm 0.09$ | **6.80** |
| TAG-TabPFNv2 (ICL) | $43.97 \pm 0.52$ | $93.11 \pm 0.17$ | $87.80 \pm 0.20$ | $15.07 \pm 1.32$ | $82.59 \pm 0.12$ | **5.40** |
| TAG-LimiX (ICL) | $44.89 \pm 0.15$ | $92.85 \pm 0.14$ | $88.09 \pm 0.18$ | $15.53 \pm 1.99$ | $82.64 \pm 0.17$ | **4.20** |
| GraphPFN (ICL) | $44.94 \pm 0.15$ | $92.42 \pm 0.21$ | $90.50 \pm 0.15$ | $21.38 \pm 0.21$ | $82.01 \pm 0.19$ | **4.00** |
| G2T-LimiX (FT) | $46.05 \pm 0.23$ | $92.53 \pm 0.13$ | $90.88 \pm 0.14$ | $20.83 \pm 0.44$ | $83.11 \pm 0.39$ | **2.60** |
| GraphPFN (FT) | $46.27 \pm 0.27$ | $94.06 \pm 0.10$ | $90.50 \pm 0.15$ | $22.15 \pm 0.42$ | $83.99 \pm 0.30$ | **1.20** |

## 5.2. Experimental Results

Tables 1 and 2 present the results of our experiments. Below, we summarize and discuss our key observations. Additional experimental results are provided in Appendix C.

> **Observation 1.** *GraphPFN shows strong ICL performance, outperforming GNNs and other GFMs on average on both collections of datasets.*

Specifically, when compared with GNNs and other GFMs evaluated in an ICL regime, GraphPFN (ICL) achieves the best average rank on both collections of datasets. Importantly, unlike G2T-FM, which also achieves strong ICL performance, GraphPFN does not require potentially heavy preprocessing like the computation of Laplacian positional encodings or PageRank.

> **Observation 2.** *Finetuning allows GraphPFN to achieve the best results on 12 out of 13 datasets, and the gains over the second-best method are often substantial.*

Specifically, the finetuned GraphPFN achieves the best performance among all considered methods on all datasets except for pubmed. Moreover, GraphPFN often yields substantial gains over the second-best method, for example, of at least one percentage point on the artnet-exp, artnet-views, avazu-ctr, city-roads-M, hm-prices, twitch-views, and questions datasets.

We hypothesize that this strong performance stems from GraphPFN's ability to capture complex graph patterns via message passing, in contrast to G2T-FM and TAG, which rely on hand-crafted graph-based features. At the same time, the pretraining procedure appears to be critical. When we replace the pretrained graph adapters with randomly initialized ones and finetune on a downstream dataset, performance drops substantially; see Appendix C for details.

## 6. Conclusion

In this work, we propose GraphPFN: a prior-data fitted graph foundation model for node-level tasks. Following the PFN framework, GraphPFN was pretrained on synthetic datasets drawn from a carefully designed prior over attributed graphs to perform predictions in the in-context learning or finetuning regime. Our experiments show strong results for GraphPFN. Even in the ICL regime, GraphPFN outperforms classic GNNs and prior ICL GFMs on average. After finetuning, GraphPFN consistently achieves state-of-the-art results, often bringing substantial improvements over the second-best method. We believe our work shows that pretraining graph foundation models on synthetic datasets drawn from a well-designed prior is a promising direction for developing truly generalizable graph foundation models. Despite promising results, the current implementation of GraphPFN has several limitations, which we discuss in Appendix A.

## Impact Statement

This paper presents work whose goal is to advance the field of Machine Learning. There are many potential societal consequences of our work, none of which we feel must be specifically highlighted here.

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

# A. Limitations and Future Work

Despite promising results, the current implementation of GraphPFN has several limitations that may stimulate future research:

- GraphPFN is difficult to scale to very large datasets since its current implementation requires processing the entire dataset at once, which leads to significant memory consumption. Developing more scalable graph foundation models can be a promising direction for future work. For example, one can utilize more memory-efficient TFMs (like TabICL (Qu et al., 2025; 2026)) as backbones or combine GraphPFN with subgraph sampling methods (Hamilton et al., 2017; Zeng et al., 2020; Zou et al., 2019; Chiang et al., 2019; Zeng et al., 2021; Balin & Çatalyürek, 2023).

- The proposed graph prior is focused on social and information networks and does not cover graphs from specific domains like traffic networks. Extending the prior with more diverse random graph models (e.g., geometric graphs (Barthélemy, 2011)) can further improve the results of GraphPFN and make its performance more robust.

- GraphPFN inherits some limitations from its tabular backbone LimiX. For example, GraphPFN does not natively support more than 10 classes in multiclass classification. This limitation can be alleviated with further development of TFMs or by using approaches such as error-correcting output codes (ECOC), as proposed in Hollmann et al. (2025). We provide experiments with ECOC in Appendix C.

- Currently, GraphPFN is limited to node-level prediction tasks and cannot handle link prediction or graph-level tasks (e.g., graph classification or regression). While we have taken a first step towards link prediction by adding a masked graph modeling head to GraphPFN during pretraining, GraphPFN still heavily relies on the presence of node-level labels to make predictions. Therefore, we cannot directly apply GraphPFN to link prediction tasks. Extending GraphPFN to other prediction tasks can be a promising direction for future research.

- Finally, we observed some variability in the overall pretraining pipeline. In our experiments, in-context learning performance varied across pretraining seeds, with standard deviations ranging from 0.3 to 0.7 percentage points. Improving the robustness of the pretraining procedure remains an important direction for future work.

# B. Graph Prior Analysis

We analyze the proposed graph prior from three complementary perspectives. First, we visually inspect the generated graph structures in Appendix B.1. Second, we provide a numerical comparison of graph characteristics to demonstrate the alignment between our synthetic prior and real-world datasets in Appendix B.2. Finally, we examine the effect of the specific graph generation procedure on downstream performance in Appendix B.3.

## B.1. Synthetic Graph Examples

We design our prior to generate graphs that are both realistic and diverse. In Figure 3, we provide examples of our synthetic graphs. Note that these graphs tend to exhibit both community structure and core-periphery structure. Our graph generation process allows us to easily control various graph properties such as their size or density.

## B.2. Numerical Analysis of Graph Characteristics

To further assess how closely our synthetic datasets match the properties of real-world graphs, we perform a numerical analysis of several graph characteristics. Specifically, we sample 10,000 synthetic datasets from the prior and compute the following statistics.

First, we report basic structural measures, including the average degree, the clustering coefficient, and an estimate of the average pairwise distance. Second, to quantify the tendency of adjacent nodes to have similar targets, we compute unbiased homophily (Mironov & Prokhorenkova, 2024) for classification datasets and assortativity (Newman, 2003) for regression datasets. Finally, to evaluate how well the graphs follow a power-law degree distribution, we compute the cumulative degree distribution: for each degree value $k$, we compute $P_k = \mathbb{P}(\deg(u) \geq k)$. We then fit a linear regression to the pairs $(\log k, \log P_k)$ and report the corresponding $R^2$.

Table 4 summarizes the distributions of these characteristics for synthetic datasets, while Table 3 reports the same statistics for real datasets. In most cases, the distribution of characteristics in the synthetic datasets covers that of the real-world datasets.

*Table 3.* Characteristics of real-world graphs from the GraphLand benchmark.

| | artnet-exp | artnet-views | avazu-ctr | city-reviews | city-roads-M | hm-prices | tolokers-2 | twitch-views |
|---|---|---|---|---|---|---|---|---|
| Average degree | 11.12 | 11.12 | 288.04 | 15.66 | 3.75 | 460.92 | 88.28 | 80.87 |
| Clustering coefficient | 0.03 | 0.03 | 0.24 | 0.26 | 0.00 | 0.27 | 0.23 | 0.02 |
| Average pairwise distance | 4.34 | 4.43 | 3.72 | 4.78 | 129.89 | 2.39 | 2.84 | 2.84 |
| Homophily | 0.28 | – | – | 0.69 | – | – | 0.10 | – |
| Assortativity | – | 0.19 | 0.18 | – | 0.74 | 0.12 | – | -0.41 |
| Degree power-law $R^2$ | 0.70 | 0.70 | 0.90 | 0.92 | 0.52 | 0.84 | 0.83 | 0.97 |

*Table 4.* Summary of distribution of characteristics of synthetic graphs from the prior.

| | Average | Std | Min | Q25 | Median | Q75 | Max |
|---|---|---|---|---|---|---|---|
| Average degree | 83.77 | 79.95 | 2.36 | 10.94 | 65.99 | 134.9 | 413.58 |
| Clustering coefficient | 0.19 | 0.14 | 0.0 | 0.06 | 0.16 | 0.28 | 0.7 |
| Average pairwise distance | 3.09 | 1.32 | 1.72 | 2.22 | 2.47 | 3.52 | 9.6 |
| Homophily | 0.06 | 0.14 | $-0.6$ | 0.0 | 0.01 | 0.08 | 0.93 |
| Assortativity | 0.06 | 0.17 | $-0.43$ | 0.0 | 0.0 | 0.07 | 0.9 |
| Degree power-law $R^2$ | 0.72 | 0.17 | 0.3 | 0.56 | 0.75 | 0.87 | 0.99 |

*Table 5.* Results of pretraining GraphPFN on simpler graph models. All models evaluated in an ICL regime without ensembling.

| | artnet-exp | city-reviews | tolokers-2 | artnet-views | avazu-ctr | city-roads-M | hm-prices | twitch-views |
|---|---|---|---|---|---|---|---|---|
| Erdős–Rényi | $48.70 \pm 0.04$ | $78.90 \pm 0.02$ | $54.69 \pm 0.27$ | $59.05 \pm 0.05$ | $20.96 \pm 0.03$ | $63.64 \pm 0.08$ | $70.61 \pm 0.05$ | $65.21 \pm 0.06$ |
| Modified Barabási–Albert | $49.49 \pm 0.06$ | $79.74 \pm 0.02$ | $58.23 \pm 0.21$ | $61.36 \pm 0.03$ | $27.43 \pm 0.08$ | $63.05 \pm 0.20$ | $70.86 \pm 0.08$ | $71.18 \pm 0.06$ |
| Degree-corrected SBM | $51.14 \pm 0.03$ | $79.85 \pm 0.02$ | $61.71 \pm 0.14$ | $62.23 \pm 0.02$ | $32.98 \pm 0.04$ | $64.30 \pm 0.08$ | $78.19 \pm 0.07$ | $73.37 \pm 0.09$ |
| Ours | $51.46 \pm 0.11$ | $80.01 \pm 0.03$ | $60.97 \pm 0.13$ | $62.07 \pm 0.05$ | $31.43 \pm 0.06$ | $64.03 \pm 0.11$ | $77.33 \pm 0.06$ | $72.75 \pm 0.04$ |

## B.3. Pretraining on Simpler Random Graphs

To assess the utility of the proposed graph generation procedure, we conduct an ablation study in which we replace it during pretraining with simpler random graph models. Specifically, we consider three alternatives: the Erdős–Rényi model (Erdős & Rényi, 1959), the degree-corrected SBM (Karrer & Newman, 2011), and a modified Barabási–Albert process (Albert & Barabási, 2002), where we treat the initial degree of a new node as a random variable and generate new nodes with different initial degrees. The results are presented in Table 5.

Our procedure yields consistently better results than both the Erdős–Rényi and modified Barabási–Albert models. Interestingly, the modified Barabási–Albert model also yields better results than the Erdős–Rényi model on all datasets except `city-roads-M`. These results might suggest that encoding relevant inductive biases in the prior improves performance.

At the same time, the degree-corrected SBM performs on par with and on some datasets better than our proposed prior. In the previous revision of this paper, which used a different pretraining procedure, a similar experiment yielded a different result, with our prior performing better than the degree-corrected SBM. The modifications of the pretraining procedure we have made in this revision lead to faster pretraining and improved predictive performance for both our graph generator and the degree-corrected SBM. Since the performance differences for these generators are relatively small, we decided to keep our proposed prior in all main experiments for consistency. Additionally, our approach generates much more diverse and realistic graphs which may potentially be useful for future studies and other application domains.

## C. Additional Results

**MGM ablation**  To assess the contribution of the masked graph modeling (MGM) component in GraphPFN, we pretrain GraphPFN without this component while keeping all other settings unchanged. The results are reported in Table 6. Removing MGM leads to consistently worse or comparable performance, indicating importance of this component for the performance of GraphPFN.

**Graph adapters architecture**  GraphPFN uses multi-head attention-based message-passing graph adapters. This design unifies the model under a single mechanism: GraphPFN uses only multi-head attention throughout, with three different masks. For completeness, we also evaluated two alternative adapters based on mean aggregation and GCN-style aggregation.

*Table 6.* Ablation results for GraphPFN. We evaluate the model without the masked graph modeling (MGM) component and replace the attention-based graph adapter with alternative architectures. All models are evaluated in an ICL setting without ensembling.

| | artnet-exp | city-reviews | tolokers-2 | artnet-views | avazu-ctr | city-roads-M | hm-prices | twitch-views |
|---|---|---|---|---|---|---|---|---|
| GraphPFN | $51.46 \pm 0.11$ | $80.01 \pm 0.03$ | $60.97 \pm 0.13$ | $62.07 \pm 0.05$ | $31.43 \pm 0.06$ | $64.03 \pm 0.11$ | $77.33 \pm 0.06$ | $72.75 \pm 0.04$ |
| w/o MGM | $50.41 \pm 0.05$ | $80.08 \pm 0.02$ | $60.59 \pm 0.14$ | $61.53 \pm 0.07$ | $31.32 \pm 0.08$ | $63.46 \pm 0.05$ | $76.10 \pm 0.12$ | $72.20 \pm 0.06$ |
| w/ GCN Adapter | $49.31 \pm 0.13$ | $79.59 \pm 0.03$ | $57.20 \pm 0.67$ | $61.61 \pm 0.03$ | $31.51 \pm 0.04$ | $61.14 \pm 0.18$ | $72.68 \pm 0.10$ | $73.07 \pm 0.03$ |
| w/ Mean Adapter | $50.03 \pm 0.13$ | $79.74 \pm 0.05$ | $58.59 \pm 0.58$ | $61.79 \pm 0.12$ | $31.29 \pm 0.12$ | $59.10 \pm 0.49$ | $74.16 \pm 0.15$ | $68.77 \pm 0.24$ |

*Table 7.* Additional experimental results on the GraphLand datasets. The suffix 'L' (Light) indicates that G2T-FM or GraphPFN was evaluated without ensembling. 'AFT' denotes adapter finetuning, while 'LimiX+GA' refers to the LimiX model equipped with randomly initialized graph adapters (GA). All other notation is consistent with the tables from the main text.

| | artnet-exp | city-reviews | tolokers-2 | artnet-views | avazu-ctr | city-roads-M | hm-prices | twitch-views | AR |
|---|---|---|---|---|---|---|---|---|---|
| LightGBM + NFA | $46.13 \pm 0.04$ | $78.53 \pm 0.01$ | $56.34 \pm 0.06$ | $56.10 \pm 0.02$ | $31.71 \pm 0.01$ | $61.18 \pm 0.03$ | $70.84 \pm 0.04$ | $60.14 \pm 0.01$ | **11.50** |
| GCN | $44.86 \pm 0.36$ | $77.81 \pm 0.15$ | $56.27 \pm 0.31$ | $56.03 \pm 0.25$ | $32.00 \pm 0.16$ | $58.82 \pm 0.25$ | $68.02 \pm 0.42$ | $75.51 \pm 0.05$ | **11.88** |
| GraphSAGE | $45.14 \pm 0.36$ | $78.17 \pm 0.10$ | $54.43 \pm 0.34$ | $49.32 \pm 0.91$ | $31.44 \pm 0.16$ | $59.44 \pm 0.27$ | $70.00 \pm 0.74$ | $66.29 \pm 0.32$ | **13.12** |
| GAT | $45.06 \pm 0.52$ | $77.74 \pm 0.21$ | $57.41 \pm 0.85$ | $53.60 \pm 0.24$ | $32.63 \pm 0.17$ | $59.86 \pm 0.20$ | $72.07 \pm 1.22$ | $72.89 \pm 0.27$ | **11.12** |
| GT | $46.41 \pm 0.71$ | $77.34 \pm 0.21$ | $56.98 \pm 0.55$ | $53.37 \pm 0.46$ | $31.11 \pm 0.49$ | $59.55 \pm 0.28$ | $69.44 \pm 0.94$ | $72.13 \pm 0.13$ | **13.00** |
| G2T-LimiX (ICL, L) | $48.44 \pm 0.23$ | $77.81 \pm 0.49$ | $61.45 \pm 0.33$ | $60.94 \pm 0.11$ | $32.39 \pm 0.13$ | $64.53 \pm 0.09$ | $74.96 \pm 0.07$ | $71.08 \pm 0.07$ | **9.12** |
| GraphPFN (ICL, L) | $51.46 \pm 0.11$ | $80.01 \pm 0.03$ | $60.97 \pm 0.13$ | $62.07 \pm 0.05$ | $31.43 \pm 0.06$ | $64.03 \pm 0.11$ | $77.33 \pm 0.06$ | $72.75 \pm 0.04$ | **8.12** |
| G2T-LimiX (ICL) | $48.52 \pm 0.29$ | $77.97 \pm 0.53$ | $61.58 \pm 0.32$ | $60.96 \pm 0.10$ | $32.41 \pm 0.14$ | $64.69 \pm 0.05$ | $74.97 \pm 0.06$ | $71.30 \pm 0.07$ | **8.12** |
| GraphPFN (ICL) | $51.79 \pm 0.11$ | $80.25 \pm 0.05$ | $61.29 \pm 0.12$ | $62.79 \pm 0.08$ | $31.63 \pm 0.06$ | $64.85 \pm 0.13$ | $77.88 \pm 0.09$ | $73.20 \pm 0.08$ | **5.88** |
| G2T-LimiX (FT, L) | $49.89 \pm 0.18$ | $80.14 \pm 0.06$ | $61.28 \pm 0.70$ | $62.08 \pm 0.12$ | $34.02 \pm 0.27$ | $65.82 \pm 0.18$ | $76.27 \pm 0.24$ | $73.69 \pm 0.35$ | **6.00** |
| GraphPFN (FT, L) | $53.33 \pm 0.23$ | $80.34 \pm 0.17$ | $62.72 \pm 0.24$ | $64.37 \pm 0.04$ | $34.89 \pm 0.16$ | $66.02 \pm 0.10$ | $80.16 \pm 0.17$ | $78.46 \pm 0.09$ | **2.88** |
| G2T-LimiX (FT) | $49.96 \pm 0.09$ | $80.15 \pm 0.07$ | $60.71 \pm 0.57$ | $62.11 \pm 0.06$ | $33.86 \pm 0.33$ | $65.96 \pm 0.11$ | $76.30 \pm 0.19$ | $73.62 \pm 0.49$ | **5.88** |
| GraphPFN (FT) | $53.49 \pm 0.81$ | $80.90 \pm 0.24$ | $62.80 \pm 0.39$ | $65.35 \pm 0.06$ | $35.07 \pm 0.34$ | $67.30 \pm 0.21$ | $81.06 \pm 0.24$ | $79.00 \pm 0.14$ | **1.50** |
| GraphPFN (AFT) | $54.65 \pm 0.13$ | $80.86 \pm 0.04$ | $61.98 \pm 0.58$ | $65.24 \pm 0.13$ | $35.08 \pm 0.12$ | $67.53 \pm 0.18$ | $81.03 \pm 0.19$ | $79.19 \pm 0.05$ | **1.62** |
| LimiX+GA (FT) | $48.06 \pm 0.87$ | $79.85 \pm 0.31$ | $51.64 \pm 3.31$ | $57.52 \pm 0.33$ | $31.38 \pm 1.48$ | $64.52 \pm 0.56$ | $73.77 \pm 3.96$ | $74.78 \pm 0.68$ | **10.12** |

Table 6 reports the results. The attention-based adapters achieve the best average performance and show clear improvements on several datasets, supporting its use in GraphPFN.

**Random graph adapters** Since GraphPFN achieves strong performance in the finetuning setting, one may hypothesize that the performance comes solely from the powerful graph adapters, but not from the pretraining procedure. To test this hypothesis, we consider the following model. We start with LimiX and add graph adapters, so the architecture exactly matches that of GraphPFN. But instead of using pretrained weights for graph adapters, we employ random weights. In order to stabilize training, we initialized the last layers in all graph adapters with zeros, ensuring that the random initialization does not break the model. After that, we finetune this model following exactly the same protocol as GraphPFN. The results of this model and a comparison with GraphPFN are presented in Table 7. One can see that using random adapters instead of pretrained ones leads to significant drops in performance, supporting the importance of pretraining for achieving the strong performance of the finetuned GraphPFN.

**Adapter-only finetune** During pretraining, only the graph adapters are updated, whereas during downstream finetuning, both the graph adapters and the LimiX backbone are updated. To determine whether the adapters themselves benefit from downstream finetuning, or whether the performance gains mainly come from adapting the LimiX backbone, we evaluate GraphPFN with adapter-only finetuning. The results are shown in Table 7. Adapter-only finetuning is comparable to full finetuning on six out of eight datasets, though noticeable differences in both directions are present on `artnet-exp` and `tolokers-2`. This suggests that the pretrained adapters are the primary driver of downstream gains.

**Evaluation without ensembling** The default evaluation setup of GraphPFN uses an ensemble over 10 forward passes. For completeness, Table 7 reports the results obtained without ensembling. Overall, ensembling yields small but consistent gains, with larger improvements on certain datasets, such as `artnet-views` and `city-roads-M`. Nevertheless, GraphPFN remains competitive without ensembling. Even in the ICL regime, it outperforms the supervised baselines on average, and also improves over G2T-FM in the ICL regime.

**ECOC** Since GraphPFN employs LimiX as a backbone, it natively supports at most 10 classes in multiclass classification. To overcome this limitation, one can employ error-correcting output codes (Dieterich & Bakiri, 1994), as proposed for

*Table 8.* Evaluation on datasets with more than 10 classes.

|  | hm-categories | lastfm-asia | roman-empire | **AR** |
|---|---|---|---|---|
| GCN | $63.53 \pm 0.25$ | $82.77 \pm 0.54$ | $68.47 \pm 0.60$ | **4.67** |
| GraphSAGE | $56.03 \pm 0.65$ | $80.54 \pm 0.24$ | $73.85 \pm 0.46$ | **4.33** |
| GAT | $67.60 \pm 0.51$ | $82.78 \pm 0.23$ | $69.58 \pm 0.64$ | **3.67** |
| GT | $69.27 \pm 0.36$ | $83.33 \pm 0.35$ | $71.49 \pm 0.80$ | **2.67** |
| GraphPFN (ICL) | $55.25 \pm 0.75$ | $85.47 \pm 0.37$ | $64.47 \pm 0.29$ | **4.67** |
| GraphPFN (FT) | $78.67 \pm 0.20$ | $88.85 \pm 0.30$ | $78.05 \pm 0.68$ | **1.00** |

*Table 9.* Wall-clock runtime comparison (in seconds). The suffix 'L' (Light) indicates that GraphPFN was evaluated without ensembling.

|  | amazon-ratings | artnet-views | city-roads-M | tolokers-2 |
|---|---|---|---|---|
| GCN | $23.93 \pm 2.97$ | $16.36 \pm 1.24$ | $27.29 \pm 3.55$ | $27.24 \pm 1.44$ |
| GraphPFN (ICL, L) | $3.46 \pm 0.07$ | $4.89 \pm 0.02$ | $2.89 \pm 0.05$ | $2.12 \pm 0.02$ |
| GraphPFN (ICL) | $16.41 \pm 0.04$ | $30.25 \pm 0.07$ | $16.22 \pm 0.02$ | $7.03 \pm 0.04$ |
| GraphPFN (FT, L) | $478.88 \pm 105.82$ | $1056.07 \pm 95.33$ | $542.72 \pm 60.55$ | $188.31 \pm 38.74$ |
| GraphPFN (FT) | $1041.00 \pm 385.78$ | $3941.79 \pm 429.49$ | $1023.22 \pm 232.98$ | $195.44 \pm 35.03$ |

PFNs by Hollmann et al. (2025). We evaluated GraphPFN with 10 ECOC estimators on three datasets: `hm-categories` (21 classes), `lastfm-asia` (18 classes), and `roman-empire` (18 classes). The results are in Table 8. ICL performance varies across datasets: on `hm-categories`, GraphPFN underperforms GNNs, while on `lastfm-asia` it already outperforms them in the ICL regime. After finetuning, however, GraphPFN substantially outperforms all GNN baselines with a noticeable margin on all three datasets.

**Runtime analysis** For completeness, we report wall-clock runtimes, in seconds, for GCN and GraphPFN (in both in-context learning and finetuning regimes, with and without ensembling for GraphPFN). For each method, we evaluated end-to-end runtime with optimal hyperparameters.[3] The results are shown in Table 9. GraphPFN in the ICL regime has the same order-of-magnitude runtime as training a GCN from scratch. Even with ensembling, its runtime is at most twice that of GCN training. In contrast, finetuning is substantially more expensive. We also note that obtaining strong GNN performance typically requires an extensive hyperparameter search, whereas ICL does not require hyperparameter tuning.

## D. Additional Details on Experimental Setup

**Datasets** Table 10 presents key statistics of the datasets used in our main experiments. Importantly, the selected datasets cover diverse domains, both homophilous and non-homophilous datasets, diverse node features (tabular features for the GraphLand datasets and text-based features for the classic datasets), and diverse tasks (both node classification and node regression).

**PCA** Since some datasets (specifically, `amazon-ratings` and `questions`) with text-embedding features have relatively high feature dimensionality, which prevented us from directly finetuning on these datasets, we applied PCA to reduce the feature dimensionality to 64 (both for in-context learning and finetuning regimes). We did not apply PCA to the `pubmed` dataset, since in our preliminary experiments applying PCA to that dataset led to significantly degraded performance. As a result, GraphPFN ran out of memory during finetuning on `pubmed`, and we therefore report its in-context learning performance for this dataset in the finetuning row.

**Ensembling** By default, we evaluate GraphPFN using an ensemble of 10 members. Ensemble members share the same model weights, but apply possibly different preprocessing schemes to the input features (e.g., column permutation) and targets (e.g., error-correcting output codes for datasets with more than 10 classes). We do not employ ensembling during the pretraining phase, but use it for both final and intermediate evaluations during finetuning on downstream tasks. For each optimization step during finetuning, we sample a single random ensemble member (i.e., one preprocessing scheme) to compute the gradient. This strategy yields the advantages of ensembling while limiting computational costs. Specifically,

---

[3]All experiments were conducted on a single NVIDIA A100 GPU. The exact machines used for different runs varied slightly, leading to approximately 10% runtime noise, which does not affect our conclusions.

*Table 10.* The key statistics of the considered graph datasets.

| name | # nodes | # edges | # features | mean degree | task | # classes | homophily | feature type |
|---|---|---|---|---|---|---|---|---|
| artnet-exp | 50,405 | 280,348 | 75 | 11.1 | cls. | 2 | no | tabular |
| artnet-views | 50,405 | 280,348 | 50 | 11.1 | reg. | – | no | tabular |
| avazu-ctr | 76,269 | 10,984,077 | 260 | 288.0 | reg. | – | no | tabular |
| city-reviews | 148,801 | 1,165,415 | 37 | 15.7 | cls. | 2 | yes | tabular |
| city-roads-M | 57,073 | 107,104 | 26 | 3.8 | reg. | – | yes | tabular |
| hm-prices | 46,563 | 10,730,995 | 41 | 460.9 | reg. | – | no | tabular |
| tolokers-2 | 11,758 | 519,000 | 16 | 88.3 | cls. | 2 | no | tabular |
| twitch-views | 168,114 | 6,797,557 | 4 | 80.9 | reg. | – | no | tabular |
| amazon-ratings | 24,492 | 93,050 | 300 | 7.6 | cls. | 5 | no | text-based |
| facebook | 22,470 | 170,823 | 128 | 15.2 | cls. | 4 | yes | text-based |
| pubmed | 19,717 | 44,324 | 500 | 4.5 | cls. | 3 | yes | text-based |
| questions | 48,921 | 153,540 | 301 | 6.3 | cls. | 2 | no | text-based |
| wiki-cs | 11,701 | 215,603 | 300 | 36.9 | cls. | 10 | yes | text-based |

*Table 11.* Empirical comparison with NodePFN.

| | artnet-exp | city-reviews | tolokers-2 | amazon-ratings | facebook | pubmed | questions | wiki-cs | AR |
|---|---|---|---|---|---|---|---|---|---|
| GCN | 44.86 ± 0.36 | 77.81 ± 0.15 | 56.27 ± 0.31 | 41.43 ± 0.49 | 91.26 ± 0.21 | 85.46 ± 0.19 | 15.42 ± 0.67 | 81.74 ± 0.21 | **3.75** |
| GraphSAGE | 45.14 ± 0.36 | 78.17 ± 0.10 | 54.43 ± 0.34 | 40.07 ± 0.53 | 91.12 ± 0.22 | 86.04 ± 0.27 | 16.55 ± 0.64 | 81.50 ± 0.27 | **3.88** |
| GAT | 45.06 ± 0.52 | 77.74 ± 0.21 | 57.41 ± 0.85 | 40.67 ± 0.55 | 92.61 ± 0.21 | 84.81 ± 0.24 | 16.75 ± 0.67 | 82.25 ± 0.27 | **3.12** |
| GT | 46.41 ± 0.71 | 77.34 ± 0.21 | 56.98 ± 0.55 | 41.56 ± 0.40 | 91.71 ± 0.22 | 84.95 ± 0.19 | 14.03 ± 0.90 | 82.54 ± 0.21 | **3.12** |
| NodePFN (ICL) | 24.19 ± 0.41 | 70.58 ± 0.06 | 43.79 ± 0.19 | 41.22 ± 0.41 | 89.42 ± 0.08 | 83.53 ± 0.11 | 7.37 ± 0.14 | 78.10 ± 0.38 | **5.75** |
| GraphPFN (ICL) | 51.79 ± 0.11 | 80.25 ± 0.05 | 61.29 ± 0.12 | 44.94 ± 0.15 | 92.42 ± 0.21 | 90.50 ± 0.15 | 21.38 ± 0.21 | 82.01 ± 0.19 | **1.38** |

because intermediate evaluations during finetuning occur only once every 10 gradient steps, evaluating the full ensemble requires just 9 additional forward passes for every 10 forward-backward passes. Therefore, the maximum time overhead of our ensembling approach with this strategy is at most double that of standard single-model finetuning.

# E. Concurrent Work

During the preparation of this paper, we became aware of a concurrent work NodePFN (Choi et al., 2026), which also pretrains a PFN-based model for graph node-level tasks. Both our paper and NodePFN were first publicly released at the same time, and therefore should be considered as concurrent works. For completeness of our study, below we describe the key differences between our works and provide their empirical comparison.

**Key differences** First, we initialize GraphPFN from the LimiX backbone, allowing GraphPFN to inherit its strong ICL capabilities and ability to handle diverse features and labels, while NodePFN is pretrained from scratch. Notably, pretraining from scratch allows NodePFN to set the maximal natively supported number of classes to 20, instead of the 10 classes used in most current TFMs, particularly in LimiX.

Second, NodePFN's architecture is based on the TabPFNv1 (Hollmann et al., 2023) architecture, which treats the whole sample as a single token and relies on either dimensionality reduction or padding to handle varying number of features. In contrast, GraphPFN is based on the LimiX architecture, which mostly resembles the architecture of TabPFNv2 that natively supports varying number of features by considering individual features (or their groups) as tokens.

Our methods also use different priors for synthetic data and incorporate graph message-passing into the model architecture in different ways.

**Empirical comparison** We evaluated NodePFN in our experimental setup using the official code released by its authors. We note that this evaluation is outside the regime recommended in the official NodePFN GitHub repository: the repository suggests applying NodePFN to datasets with at most 1024 training samples, whereas all datasets in our benchmark exceed this threshold. For hyperparameter selection, we used the search space proposed by the authors for the number of components in truncated SVD and the number of smoothing steps. When possible, we also included the option of not applying truncated SVD. We performed a grid search, selected the best configuration according to the validation metric, and then reported the corresponding test score. Table 11 summarizes the results, showing that GraphPFN strongly outperforms NodePFN on all the considered datasets. Since the official implementation of NodePFN lacks support for finetuning, we restrict our evaluation to the ICL setting.

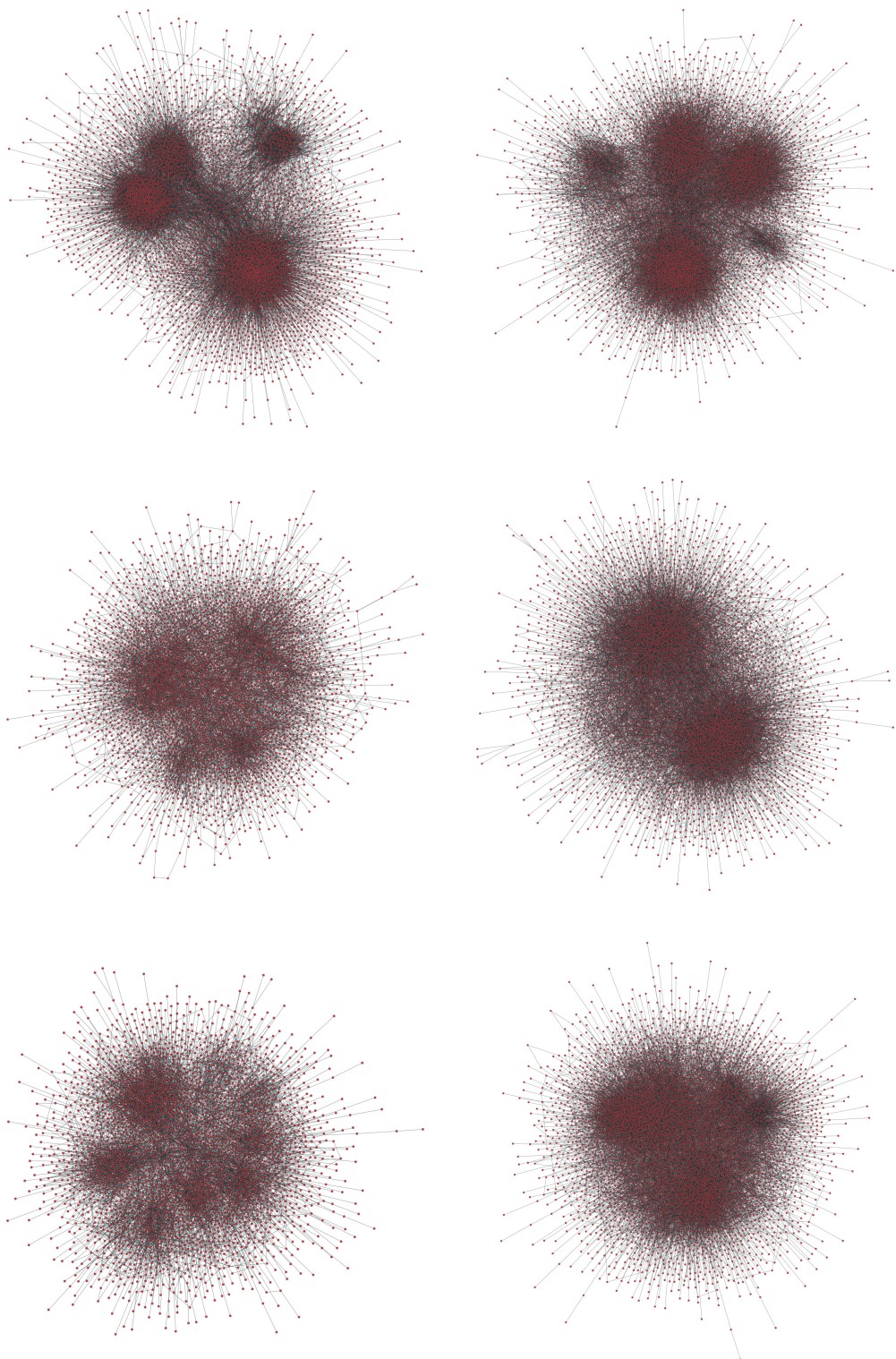

*Figure 3.* Example visualizations of graphs from our prior.

