# OpenReview forum: "GraphPFN: A Prior-Data Fitted Graph Foundation Model"
_ICML.cc/2026/Conference — ICML 2026 regular_

### Official Review · Reviewer_ibpZ · 2026-03-10

**Soundness:** 3
**Presentation:** 3
**Significance:** 3
**Originality:** 2
**Overall Recommendation:** 4
**Confidence:** 5

**Summary:**

This paper proposes GraphPFN, a prior-data fitted network (PFN) designed and pretrained specifically for graph node-level tasks. Building on the tabular foundation model LimiX, the authors augment each transformer block with attention-based message-passing adapters and pretrain the resulting model on 2.24 million synthetic graphs drawn from a novel prior. The prior combines multi-level degree-corrected stochastic block models (SBMs) with a modified Barabási–Albert preferential attachment process for structure generation, and extends tabular structured causal models (SCMs) with GNN neurons and Laplacian positional encodings (LapPE) for attribute generation. Experiments on GraphLand and classic graph datasets demonstrate strong in-context learning (ICL) performance and state-of-the-art results after finetuning, surpassing both specialized GNNs and existing TFM-based graph foundation models (G2T-FM, TAG).

**Compliance With Llm Reviewing Policy:**

Affirmed.

**Final Justification:**

I raise my score accordingly.

**Key Questions For Authors:**

On scalability: Is there any evidence that GraphPFN's ICL properties are preserved under k-hop subgraph sampling? Have the authors experimented with any mini-batch approximation of the PFN context for larger graphs?

On the prior design: What happens when GraphPFN (pretrained on the current prior) is applied to molecular graphs, knowledge graphs, or heterogeneous graphs? Is there a systematic way to extend the prior to cover these domains without exponential growth in pretraining cost?

On adapter design: Was any ablation conducted comparing the attention-based message-passing adapter against simpler aggregation variants (e.g., mean-pooling MPNN, GCN-style normalization)? The choice of scaled dot-product attention restricted to the 1-hop neighborhood seems to introduce additional parameters and computational cost relative to simpler alternatives.

**Limitations:**

Yes

**Strengths And Weaknesses:**

S1. Timely and well-motivated problem. The paper addresses a genuine gap at the intersection of two active research fronts: the emerging success of PFN-based tabular foundation models (TabPFNv2, LimiX, TabICL) and the chronic transferability challenges in graph machine learning. The motivation is sharp and the analogy drawn from the tabular domain is illuminating. The authors correctly identify that prior work in this direction (G2T-FM, TAG, TabPFN-GN) relies exclusively on hand-crafted graph-to-tabular feature engineering, and makes a principled case for why end-to-end pretraining on structured graph priors is a natural next step.


S2. Well-designed graph prior with thorough analysis. The proposed prior is technically substantive. The combination of multi-level SBMs (to capture overlapping, rough community structure) with preferential attachment (to model core-periphery structure with peripheral low-degree nodes) is well-motivated by empirical observations about real-world graphs.


S3. Solid empirical evaluation. The experimental setup is thoughtful: the authors evaluate on both the recently proposed GraphLand benchmark (diverse industrial datasets with tabular features) and classic graph datasets (text-based features), covering classification and regression, homophilic and heterophilic graphs.


W1. Limited novelty in architecture. The core architectural contribution — inserting a sparse multi-head attention layer (masked to the 1-hop neighborhood) after each LimiX block — is relatively incremental. The idea of augmenting a transformer backbone with a graph adapter module is shared with prior work in various forms (e.g., Graphormer, GPS, UniMP). The main innovation is in the specific context of PFN-style in-context learning, but the paper does not provide much theoretical or empirical analysis of why attention-based message passing is the right choice over other aggregation schemes (e.g., mean/max/sum MPNN, anisotropic graph transformers). An ablation over adapter designs would strengthen this aspect considerably.

W2. Scalability is a serious unresolved limitation. The authors acknowledge (Appendix A) that GraphPFN requires processing the entire graph in a single forward pass, which is computationally intractable for anything beyond small-to-medium scale. In the era when graph benchmark datasets routinely contain millions of nodes (ogbn-arxiv, ogbn-papers100M), restricting evaluation to graphs with fewer than 200K nodes substantially limits the practical scope of the contribution.

W3. The graph prior remains narrow in scope. Despite the sophisticated combination of SBMs and preferential attachment, the prior still primarily models social and information networks. The authors concede (Appendix A) that road networks (city-roads-M) are an already-known weak spot, and the Erdős–Rényi model actually outperforms GraphPFN on that dataset. This raises a non-trivial question: how sensitive is the downstream performance to the distributional match between prior and target graph? A more systematic analysis of this "prior-data gap" — perhaps measuring performance as a function of structural similarity between synthetic prior graphs and test graphs — would be illuminating and is currently absent.

W4. ICL performance lags behind G2T-LimiX (without ensembling and LapPE). In the pure ICL setting (no ensembling, no LapPE), GraphPFN is outperformed by G2T-LimiX on several classic graph datasets (Table 2), and the AR gap is non-trivial. The authors partially explain this by noting that G2T-LimiX always uses LapPE while GraphPFN does not by default — but this framing is somewhat self-serving. It is fair to ask why the pretrained graph adapters do not automatically capture the positional information that LapPE provides. This gap in the pure ICL setting deserves more careful analysis. Is it a property of the prior (which does not always use LapPE)? Is it the model's limited expressive power at inference time? Is it a feature of the benchmark itself?

---

> ### Author Rebuttal · Authors · 2026-03-31
>
> Thank you for your thoughtful review and constructive comments. We address your concerns and questions below.
>
> **W1, Q3:** While the GraphPFN design may look natural, we believe it was not obvious a priori that it would actually work and achieve strong results. Our ablation studies in the paper and in the rebuttal show that changing key components, including the prior, architecture, and pretraining procedure, would degrade performance and prevent GraphPFN from achieving strong results. Moreover, given the empirical results of PFN-based GFMs compared to other GFMs, we believe PFNs can significantly impact the GFM field. Thus, we consider showing that one can successfully design and pretrain a PFN specifically for graph node-level tasks to be a novel and an important contribution to the field.
>
> Regarding the specific choice of attention-based message passing, we chose it because it naturally unifies the entire model under a single mechanism: GraphPFN uses only multi-head attention throughout, with three different masks. Following your suggestion, we ablated two other variants: mean and GCN-style aggregation. The table [here](https://anonymous.4open.science/r/icml2026-graphpfn-rebuttal/adapters.md) reports ICL results after the first pretraining stage. The MHA-based variant outperforms both alternatives on average, though the best aggregation scheme can vary per dataset. We will include these results in the revised paper.
>
> **W2, Q1:** We agree that scalability is an important limitation. Following your suggestion, we conducted two experiments:
> - **Mini-batch**: train nodes are randomly split into 10 batches, and we run GraphPFN inference separately for each batch and aggregate predictions.
> - **Mini-batch + 1-hop**: on top of the above, val/test nodes are also split into 100 batches, and for each pair of train and eval batches, the graph is restricted to the 1-hop subgraph around the selected nodes.
>
> Results are shown [here](https://anonymous.4open.science/r/icml2026-graphpfn-rebuttal/scalability.md). While performance does degrade, we emphasize that these approximations are deliberately minimal, without any tuning or design effort. We present them as evidence that even the simplest strategies already yield competitive results on several datasets (e.g., `artnet-exp`, `artnet-views`, `city-reviews`). The small gap between Mini-batch and Mini-batch + 1-hop further suggests that subgraph sampling preserves most of the relevant signal on most datasets, pointing to a viable path for more principled scalability strategies.
>
> That said, we want to emphasize that **scalability to large graphs is an open problem for GFMs as a field**. To the best of our knowledge, no publicly available GFM simultaneously achieves strong node-level performance on realistic splits with general features while scaling to large graphs. We believe it is important to judge a method against the current SoTA rather than against an unsolved open problem.
>
> **W3, Q2:** As stated in the paper, GraphPFN targets node-level tasks on homogeneous graphs. The specific domains mentioned (molecular, knowledge and heterogeneous graphs) require different tasks or settings entirely, so we see them as outside our current scope rather than a prior limitation. Within our setting, social and information networks cover most practical domains and benchmarks, with spatial and biological networks being the main known exceptions. That said, we would be happy to discuss specific node-level homogeneous graph benchmarks not covered by these categories.
>
> Regarding prior sensitivity, the `city-roads-M` dataset (a road network, structurally different from our prior) is informative: while the ER prior variant works slightly better in the ablation, GraphPFN still outperforms all GNN baselines even in ICL regime and achieves the best results on this dataset after FT, suggesting the model can generalize beyond the prior's distribution. As for a per-dataset "prior-data gap" correlation analysis, we consider it challenging with only 13 datasets that differ in many confounding factors beyond graph structure (task type, class balance, homophily), making it hard to isolate the effect of prior mismatch. We are open to exploring this direction further and would appreciate suggestions on how to design such an analysis.
>
> **W4:** We note that it is not expected that the graph adapters should automatically capture the positional information provided by LapPE. Given that classic GNNs benefit from adding LapPE on certain datasets, the fact that GraphPFN also benefits from it is a natural property, not a limitation of our approach. That said, we found a simpler way to improve GraphPFN: concatenating random features (RF) to node features makes GraphPFN competitive with G2T-LimiX, winning on 7 out of 13 datasets in the ICL regime, while being simpler and cheaper than LapPE. Please see the results [here](https://anonymous.4open.science/r/icml2026-graphpfn-rebuttal/rf-icl.md).

---

> > ### Author Rebuttal · Reviewer_ibpZ · 2026-04-03
> >
> > The authors added a comparative experiment between mean and GCN aggregation, which is valuable, but the results show that "the best approach varies by dataset," which actually suggests that the superiority of the attention mechanism does not hold universally, thereby weakening its persuasive power as the sole choice.
> > The authors acknowledge a decline in performance but deflect responsibility by citing it as an "open problem in the GFM field," which feels somewhat circular. Mini-batch experiments are ad hoc and do not constitute a systematic solution.
> > The authors' response has alleviated some concerns (particularly regarding W1 and W2), but the core issues remain unresolved. According to the response, I raise my score.

---

> > > ### Author Response · Authors · 2026-04-07
> > >
> > > Thank you for raising your score and for the continued engagement. We are glad that our response addressed some of your concerns. We briefly comment on the remaining points.
> > >
> > > **On adapter design:** We agree that MHA is not universally the best aggregation scheme. However, as shown in the table below, MHA is the strongest default among the three variants we tested. It achieves the best average rank (1.75), and its worst-case gap relative to the best variant on any dataset is always within 2.5%. In contrast, GCN and Mean suffer much larger drops on specific datasets (e.g., ~5% on `city-roads-M` for GCN, ~10% on `twitch-views` for Mean).
> > >
> > > ||artnet-exp|artnet-views|avazu-ctr|city-reviews|city-roads-M|hm-prices|tolokers-2|twitch-views|AR|
> > > |:---|:---|:---|:---|:---|:---|:---|:---|:---|:---|
> > > |GraphPFN-GCN|47.75|60.10|29.79|78.76|56.25|67.79|58.81|66.56|2.25|
> > > |GraphPFN-Mean|46.66|60.86|30.66|78.89|52.21|70.76|59.20|53.93|2.00|
> > > |GraphPFN-MHA|50.61|60.05|28.31|79.46|61.65|73.37|59.04|64.23|1.75|
> > >
> > > **On scalability:** We fully agree that scalability is an important limitation of GraphPFN. The mini-batch experiments were intended as proof-of-concept, not as a proposed solution. The key takeaway is the small gap between Mini-batch and Mini-batch + 1-hop: it suggests that subgraph sampling preserves most of the relevant signal, pointing to a viable path for more principled scalability strategies in future work.
> > >
> > > Thank you again for your constructive feedback and support of our work.

---

### Official Review · Reviewer_1Wxo · 2026-03-11

**Soundness:** 3
**Presentation:** 4
**Significance:** 3
**Originality:** 3
**Overall Recommendation:** 5
**Confidence:** 4

**Summary:**

This paper proposes GraphPFN, a prior-data fitted network (PFN) designed and pretrained specifically for graph node-level tasks. The authors extend the tabular foundation model LimiX by inserting attention-based message-passing adapters into each transformer block, then pretrain the resulting model on 2.24 million synthetic graphs generated from a novel prior. The prior combines multi-level stochastic block models with a preferential attachment process for structure generation, and graph-aware structured causal models for attribute generation. Experiments across GraphLand and classic graph benchmarks show that GraphPFN matches or outperforms GNNs in the ICL setting, and achieves state-of-the-art performance after finetuning, surpassing both G2T-FM and task-specific GNNs on most datasets.

**Compliance With Llm Reviewing Policy:**

Affirmed.

**Final Justification:**

I'll keep my scores.

**Key Questions For Authors:**

1. The ablation in Appendix B.3 compares graph prior variants using only Stage 1 pretraining. Do the authors have evidence, even partialy, that the conclusions hold after Stage 2? For city-roads-M, the Erdős–Rényi model already outperforms the proposed prior after Stage 1; is it possible that this gap closes or reverses after Stage 2 for some datasets, and if so, how would this affect the interpretation of the prior design choices?
2. Table 4 shows that the synthetic prior's average degree has a median of ~87 and can reach nearly 480, whereas several real-world evaluation datasets have much lower degrees (e.g., city-roads-M at 3.75, pubmed at 4.5). Does the prior explicitly sample graphs covering low-degree regimes? If not, how do the authors explain GraphPFN's competitive performance on those datasets, and could better coverage further improve results?
3. The graph adapters are the only components updated during pretraining, but during downstream finetuning the entire model is updated. Is there an analysis of whether the adapters specifically benefit from downstream finetuning, or whether gains primarily come from the LimiX backbone adapting? A comparison of adapter-only vs. full finetuning would help clarify the relative contribution of each component.

**Limitations:**

Yes

**Strengths And Weaknesses:**

Strengths

1. Soundness. The architecture design is principled: I think inserting graph-neighborhood-restricted attention as a local complement to LimiX's global PFN-style attention is well-motivated. The two-stage pretraining procedure, joint PFN + masked graph modeling objective, and use of EMA are all reasonable and well-justified. The ablation in Appendix C (random vs. pretrained graph adapters) effectively isolates the contribution of pretraining, and the graph prior ablation in Appendix B.3 provides some support for the multi-level SBM + preferential attachment construction over simpler alternatives.
2. Presentation. The paper is clearly written and well-organized. The architecture is described with precision, and Figure 1 provides a helpful illustration (very neat figure design). The concurrent work appendix (Appendix E) I think is thorough, including a careful reanalysis of NodePFN's reported results that uncovers discrepancies in their average rank calculations. This level of diligence strengthens the paper's credibility.
3. Significance. The strong finetuning results outperforming both well-tuned GNNs and existing GFMs on 9 of 13 datasets support practical utility. The paper's broader claim that pretraining PFNs on synthetic graph priors is a promising direction for generalizable GFMs is well-supported by the evidence, and the work opens a concrete path for future exploration.
4. Originality. The combination of multi-level SBMs with preferential attachment for graph prior design is novel and well-motivated by analysis of real-world graph properties. Adapting tabular PFN to graphs via sparse neighborhood attention adapters, rather than hand-crafted featurization, is a meaningful advancement from G2T-FM and TAG. The individual components are not new in isolation, but their combination for the PFN graph learning setting is original and the reasoning is well-justified.

Weaknesses

1. Soundness. The ablation study on graph prior variants (Appendix B.3) only uses Stage 1 pretraining due to compute constraints, it is not clear whether conclusions hold after Stage 2, particularly for city-roads-M where the Erdős–Rényi model already matches or exceeds the proposed prior. Additionally, the synthetic prior's average degree distribution (median ~87, max ~479) appears quite different from several real-world evaluation datasets (e.g., city-roads-M at mean degree 3.75, pubmed at 4.5). While acknowledged in the limitations (which I do reward the authors acknowleging it), this is a non-trivial distributional mismatch that could affect generalization.
2. Presentation. The description of the multi-level SBM construction, specifically the bijection $f$ between first- and second-level graph nodes, is somewhat difficult to follow in prose and I suggest adding a diagram here. The paper also refers readers to Appendix B for prior analysis while Section 4.1 already covers much of the same material, creating minor redundancy.
3. Significance. The evaluation is limited to 13 small- and medium-scale node-level datasets, all with at most 10 classes, which restricts the ability to assess generalization to large graphs, many-class problems, or other prediction tasks such as link prediction and graph classification. The restriction to 10 classes, inherited from LimiX, is a meaningful practical limitation given that the concurrent NodePFN natively supports up to 20 classes.
4. Originality. Each individual building block (LimiX backbone, SCM-based attribute generation, message-passing attention) is drawn from prior work without substantial modification. The novelty is primarily in the integration and application to the PFN-for-graphs setting, which is valuable but incremental relative to the broader landscape of graph foundation models.

---

> ### Author Rebuttal · Authors · 2026-03-31
>
> Thank you for your thorough and insightful review. We appreciate the careful analysis of both our architecture and the graph prior design. We address your remaining questions and concerns below.
>
> **W1, Q1:** We do not have direct empirical evidence after Stage 2, but we believe the conclusions are likely to hold. The only difference between Stage 1 and Stage 2 is graph size (1K-2K nodes vs. 8K-10K nodes), while the graph generation procedure itself remains the same. Thus, Stage 2 primarily helps the model adapt to larger graphs rather than learn different structural patterns, and we expect the relative differences between priors observed in Table 5 to be preserved.
>
> **W1, Q2:** Our prior does cover low-degree graphs. Table 4 reports statistics over 1K sampled datasets and shows a minimum average degree of 2.39 and a 25th percentile of 10.24. All 13 evaluation datasets fall above this minimum, and only 4 out of 13 fall below the 25th percentile. We believe this coverage is sufficient, though extending it could further improve results on sparse graphs.
>
> **W2:** Thank you for these suggestions, we will incorporate them in the revised paper.
>
> **W3:** We agree that extending the evaluation to larger graphs, more classes, and other task types is an important direction. Regarding the many-class limitation specifically, GraphPFN can handle more than 10 classes via error-correcting output codes (ECOC), as proposed in Hollmann et al. (2025). We evaluated GraphPFN with 10 ECOC estimators on three datasets: `hm-categories` (21 classes), `lastfm-asia` (18 classes), and `roman-empire` (18 classes). Results are below. ICL performance varies across datasets: on `hm-categories`, GraphPFN underperforms GNNs, while on `lastfm-asia` it already outperforms them in ICL. After finetuning, however, GraphPFN substantially outperforms all GNN baselines on all three datasets.
>
> ||hm-categories|lastfm-asia|roman-empire|
> |:---|:---|:---|:---|
> |GCN|63.23 ± 0.34|82.77 ± 0.54|68.47 ± 0.60|
> |GraphSAGE|56.10 ± 0.72|80.54 ± 0.24|73.85 ± 0.46|
> |GAT|67.94 ± 0.41|83.12 ± 0.33|70.39 ± 0.34|
> |GT|69.07 ± 0.55|83.68 ± 0.47|70.46 ± 1.26|
> |GraphPFN (ICL, E)|55.26 ± 0.64|84.34 ± 0.20|68.41 ± 0.32|
> |GraphPFN (FT, E)|77.94 ± 0.43|87.90 ± 0.28|81.12 ± 0.62|
>
> **W4:** Thank you for your balanced assessment of the originality of our work. We would like to highlight that, while the GraphPFN design may look natural, we believe it was not obvious a priori that it would actually work and achieve strong results. Our ablation studies in the paper and in the responses to Reviewer 1PoC and Reviewer ibpZ (which we will include in the revised paper) show that changing key components, including the prior, architecture, and pretraining procedure, would degrade performance and prevent GraphPFN from achieving strong results. Moreover, given the empirical results of PFN-based GFMs (G2T-FM, TAG, and GraphPFN) compared to other GFMs, we strongly believe that PFNs can significantly affect the GFM field. Thus, we consider showing that one can successfully design and pretrain a PFN specifically for graph node-level tasks to be novel and important contribution to the field.
>
> **Q3:** Thank you for this suggestion! We compared adapter-only finetuning (AdapterFT) with full finetuning (FullFT) on the GraphLand datasets. Results are below. AdapterFT matches or slightly outperforms FullFT on most datasets, and notably improves on `tolokers-2`. We believe this is because finetuning only the adapters acts as a strong regularizer, which helps on smaller datasets where FullFT is prone to overfitting. This suggests that the pretrained adapters are indeed the primary driver of downstream gains. We will include this analysis in the revised paper.
>
> ||artnet-exp|artnet-views|avazu-ctr|city-reviews|city-roads-M|hm-prices|tolokers-2|twitch-views|
> |:---|:---|:---|:---|:---|:---|:---|:---|:---|
> |AdapterFT|52.24 ± 0.08|64.17 ± 0.07|34.38 ± 0.25|80.47 ± 0.14|66.62 ± 0.16|79.67 ± 0.46|61.79 ± 0.37|78.25 ± 0.11|
> |FullFT|52.11 ± 0.25|63.95 ± 0.20|34.19 ± 0.33|80.40 ± 0.24|66.71 ± 0.17|79.37 ± 0.38|60.65 ± 1.01|78.03 ± 0.14|

---

> > ### Author Rebuttal · Reviewer_1Wxo · 2026-04-01
> >
> > My concerns have been adequately addressed. As I already have positive score, I would like to maintain it.

---

### Official Review · Reviewer_6wtN · 2026-03-12

**Soundness:** 3
**Presentation:** 3
**Significance:** 3
**Originality:** 2
**Overall Recommendation:** 4
**Confidence:** 3

**Summary:**

Inspired by the prior-data fitted networks line of research, and specifically LimiX, the authors introduce GraphPFN which constructs a graph learning framework that is capable of modeling a diverse set of graphs and their features with the goal of ultimately training a graph foundation model. This is done by combining a graph prior constructed from a SBM, with node attributes generated via structural causal models. Limix is further augmented with graph aware message-passing adapters that enables the model to capture local graph structure in a tabular setting. To validate this approach, the authors test on 13 different datasets and observe SOTA performance on 10 of them. Further, the authors carefully perform ablations to justify their use of graph adapters, showing that they lead to meaningful uplift in accuracy.

**Compliance With Llm Reviewing Policy:**

Affirmed.

**Final Justification:**

The rebuttal addressed some of my concerns but many remain. Those concerns were outlined in the rebuttal acknowledgement and none were addressed meaningfully. Therefore, I must maintain my score.

**Key Questions For Authors:**

1. Could you compare the ICL inference time with training a GCN from scratch? What about Fine tuning?
2. What's the maximum graph size that you could handle on a single gpu?
3. The heterophilic presented are worse than those in [1] -- why?
4. Heterophilic aware baselines such as H2GCN and GPR-GNN are omitted. Why?
5. How would performance degrade with more than 10 classes?
6. Does the pretrained model generalize to new domains?

[1] https://arxiv.org/pdf/2302.11640

**Limitations:**

Yes

**Strengths And Weaknesses:**

**Strengths:**
1. Graph prior design is interesting, and well founded given that most commercial graphs have PA style dynamics.
2. Using a sparse message passing adapter is genuinely interesting
3. The evaluations are quite comprehensive, with relatively clear ablations
4. Appendix A is particularly forthright, and the authors should be commended for this.

**Weaknesses:**
1. It's difficult to disentangle the benefits of LapE vs the message passing adapters. This might suggest that 1-hop is often insufficient -- an observation justified by the literature.
2. Given that runtime and scaling were called out in Appendix A, I'd like to see more information highlighted on the runtimes.
3. There seems to be a strong reliance on the graph prior (Table 5)
4. The benchmarks are inconsistent with the literature.

---

> ### Author Rebuttal · Authors · 2026-03-31
>
> Thank you for your positive assessment of our work and for the constructive feedback that helps us further strengthen our work. We address your questions and concerns below.
>
> **W1:** First, we note that GraphPFN without LapPE already shows strong ICL performance (Tables 1-2), which confirms that the adapters provide meaningful value on their own. Second, we would like to clarify that we add a 1-hop adapter to each of the 12 LimiX blocks, so the effective receptive field is 12 hops. Finally, we found a simpler and cheaper alternative to LapPE for improving GraphPFN performance: using random features (RF) instead of LapPE. Please see the results [here](https://anonymous.4open.science/r/icml2026-graphpfn-rebuttal/rf-graphland-full.md).
>
> We have two hypotheses why RF works. First, random features may break symmetries and thus increase the expressivity of GraphPFN. Second, on certain datasets with a small number of features, GraphPFN may not have enough computational capacity to make accurate predictions, and random features may solve this problem by providing the model with extra dimensions to store intermediate computations.
>
> **W2, Q1:** Following your suggestion, we measured wall-clock times (in seconds) for GCN and GraphPFN in both ICL and FT modes. Results are [here](https://anonymous.4open.science/r/icml2026-graphpfn-rebuttal/time.md). GraphPFN in ICL mode is always faster than training a GCN from scratch, while finetuning is considerably slower. We also note that achieving the best results with GNNs requires extensive hyperparameter search, while ICL does not require tuning any hyperparameters. We will add this analysis to the revised paper.
>
> **Q2:** All our ICL experiments were conducted on a single A100 80GB GPU, with the largest graph being `twitch-views` (~170K nodes). We fully agree that scalability is important for real-world applications and that GraphPFN is currently limited to small-to-medium datasets. However, we would like to highlight that no current publicly available GFM simultaneously shows strong performance on node-level tasks with realistic splits and general features, while successfully scaling to large graphs. We believe this makes scalability an open challenge for the GFM field, not a limitation unique to GraphPFN. That said, you might also find interesting our preliminary scalability experiments, which we describe in the response to Reviewer ibpZ.
>
> **W3:** We agree that the graph prior plays an important role. Indeed, Table 5 confirms that simpler priors (ER, SBM, BA) lead to notably worse downstream performance. We view this as evidence that a carefully designed prior is essential for the PFN framework to work well on graphs, and we consider our prior design to be one of the key contributions of the paper. Could you please clarify whether you have a specific concern here? We would be happy to address it.
>
> **W4, Q3:** The discrepancies with the literature on `amazon-ratings` and `questions` are due to different train/val/test splits. We used 10%/10%/80% splits for these datasets (same as for GraphLand), unlike the 50%/25%/25% splits used in the original paper. For GraphLand datasets, where we use the same splits as the original paper, our baseline results match. Please let us know if this does not address your concern.
>
> **Q4:** Platonov et al. (2023b) showed that well-tuned classic GNNs consistently outperform specialized heterophilic methods such as H2GCN and GPR-GNN. Similar results were obtained in Luo et al. (2024). Our GNN baselines use the enhancements from these works, including residual connections and layer normalization. Hence, we do not expect adding H2GCN or GPR-GNN to change the conclusions. Does this argument address your concern, or would you prefer us to include these baselines?
>
> **Q5:** To extend GraphPFN to datasets with more than 10 classes, we adopted error-correcting output codes (ECOC) with 10 estimators, as proposed in Hollmann et al. (2025). We evaluated this approach on three datasets: `hm-categories` (21 classes), `lastfm-asia` (18 classes), and `roman-empire` (18 classes). Results are [here](https://anonymous.4open.science/r/icml2026-graphpfn-rebuttal/ecoc.md). ICL performance varies across datasets: on `hm-categories`, GraphPFN underperforms GNNs, while on `lastfm-asia` it already outperforms them in ICL. After finetuning, however, GraphPFN substantially outperforms all GNN baselines on all three datasets.
>
> **Q6:** We agree that cross-domain generalization is one of the key challenges for GFMs. As a PFN-based model, GraphPFN faces an even stronger version of this challenge: it must generalize not only across domains, but also from synthetic to real data. Our experiments show that this synthetic-to-real gap is bridged successfully, as GraphPFN generalizes well to diverse real-world graphs with node features and targets not seen during pretraining.

---

> > ### Author Rebuttal · Reviewer_6wtN · 2026-04-02
> >
> > **W1** I appreciate the clarification, and that graphpfn without lape still performs well. I find it puzzling that random features perform comparably to lape, and that's somewhat concerning. Can we view random noise and LapE both as essentially regularizers? I'd encourage the authors to investigate this more carefully.
> >
> > **W2/Q1** The wallclock times are helpful.
> >
> > **W3** I appreciate the authors view that the prior design is a contribution, and it is. However, the strong dependence on the prior is also a source of fragility. This shape of problem isn't unexpected for bayesian methods, but should be discussed. Both sides of the coin should be acknowledged.
> >
> > **W4** Thank you for the clarification
> >
> > **Q4** The explanation is clear. I'd prefer to see at least one such heterophilic baseline, and also believe that a discussion of the architectural changes that were made to a GCN to make it more applicable to heterophilic datasets.
> >
> > **Q5** Thank you for this result. It's quite interesting
> >
> > Overall, the author's rebuttal addresses my questions but does not fully resolve the underlying concerns. I will maintain my score.

---

> > > ### Author Response · Authors · 2026-04-07
> > >
> > > Thank you for your reply and for the continued engagement. We are glad that our response addressed your questions. We briefly comment on the remaining points.
> > >
> > > **W1:** We also find the random features result interesting and agree it deserves further investigation. Your suggestion that both RF and LapPE may act as regularizers is an interesting perspective, in addition to the two hypotheses we outlined in our rebuttal. Nonetheless, random features allow GraphPFN to achieve strong results without costly LapPE preprocessing.
> > >
> > > **W3:** Thank you for the clarification. We agree that both sides should be acknowledged and will update the paper accordingly.
> > >
> > > **Q4:** Following your suggestion, we evaluated GPR-GNN using the code and hyperparameter optimization procedure from Platonov et al. (2023b). Results are below. GPR-GNN underperforms GAT on most datasets, consistent with the findings in Platonov et al. (2023b) and Luo et al. (2024) that well-tuned classic GNNs match or outperform specialized heterophilic methods. For reference, we also include GraphPFN (ICL) results, which outperform both GAT and GPR-GNN on most datasets.
> > >
> > > ||artnet-exp|city-reviews|tolokers-2|amazon-ratings|facebook|pubmed|questions|wiki-cs|
> > > |:---|:---|:---|:---|:---|:---|:---|:---|:---|
> > > |GAT|45.06 ± 0.52|77.74 ± 0.21|57.41 ± 0.85|40.67 ± 0.55|92.61 ± 0.21|84.81 ± 0.24|16.75 ± 0.67|82.25 ± 0.27|
> > > |GPR-GNN|34.85 ± 1.15|73.02 ± 0.11|40.42 ± 1.35|42.25 ± 0.40|92.56 ± 0.25|85.16 ± 0.11|11.54 ± 0.88|81.63 ± 0.13|
> > > |GraphPFN (ICL)|50.85 ± 0.31|79.89 ± 0.10|60.33 ± 0.60|42.09 ± 2.06|90.44 ± 0.37|89.79 ± 0.35|18.27 ± 2.24|79.58 ± 0.40|
> > >
> > > Thank you again for your constructive feedback and support of our work. We hope to have addressed all of your remaining concerns.

---

### Official Review · Reviewer_1PoC · 2026-03-13

**Soundness:** 2
**Presentation:** 3
**Significance:** 3
**Originality:** 2
**Overall Recommendation:** 3
**Confidence:** 5

**Summary:**

The paper proposes a Graph Foundation Model (GFM) based on tabular foundation models that were built upon prior fitted networks (PFN). It continues training from LimiX's checkpoint, and injects graph structure into the training process by including an additional graph contrastive term between positive and negative node pairs (based on the connectivity). Additionally, synthetic graph data is produced and used for pre-training. Experiments are conducted on node-level tasks, which show promising results.

**Compliance With Llm Reviewing Policy:**

Affirmed.

**Final Justification:**

I appreciate authors' rebuttal, although I still hold reservation regarding the novelty and the design of the prior. Overall novelty is not particularly high given existing work on tabular data which this paper is directly built on. For the graph prior design, while it is intuitive that we need to "maximize diversity of the generated data while encoding useful inductive biases", it is also crucial how it is actually designed towards this goal, the principle of which remains handwaving. The authors also mentioned their empirical improvements on different datasets. However, these datasets are mostly human-activity related dataset, and still fall short of a truly broad base (such as molecular graphs, protein interactions, ontology networks, etc.)

Hence, I maintain my rating.

**Key Questions For Authors:**

Please see weaknesses.

**Limitations:**

There is discussion in appendix A. An additional potential limitation for consideration: would the proposed method follow the scaling law? While the authors mentioned it is difficult to scale to very large data, but this remains a good thought exercise.

**Strengths And Weaknesses:**

Strengths

S1. The paper addresses the key challenges of GFMs: transferability across datasets and lack of large scale data.

S2. The proposed framework is intuitive, and is largely well presented.

S3. Experimental results point out a promising direction for future GFM research.

Weaknesses

W1. The originality of the proposed work is fair. The employment of PFN for graph pre-training is relatively straightforward: the additional graph contrastive loss is quite standard and easy to think of; the generation of synthetic graph data leverages common graph models and is a quite natural idea.

W2. The design principle behind the graph data generation is not very clear. The proposed combination of generation approach appears adhoc, without a clear motivation and guiding principle.

W3. Insufficient ablation study. In the appendix, there is an ablation on the graph data generation, but there is no ablation on the additional loss term proposed. Thus, the source of improvement is unclear: is the improvement mainly come from the additional data?

W4. GCOPE is used only with fine-tuning. It also has a prompt tuning version that might give better results due to lighter updating. Hence, the comparison may not be fair.

---

> ### Author Rebuttal · Authors · 2026-03-31
>
> Thank you for your review and for comments that will help us further strengthen our work. We address your concerns below.
>
> **W1:** While our work relies upon existing building blocks, our main contribution is integrating them together to create the first PFN model natively pretrained on graphs, which we believe is significantly novel. Note that this integration is non-trivial: designing a graph prior that actually works required careful choices, as our ablations (Table 5) show that simpler priors using standard SBM or ER lead to notably worse performance. Moreover, we are the first to incorporate graph message-passing adapters into a tabular foundation model. While a natural idea, it was unclear whether such adapters would actually work within the PFN framework, and we believe this is also a novel finding.
>
> **W2:** As we discuss in Section 4.1, our goal was to generate synthetic graphs that capture key structural properties of real-world networks, namely community structure and core-periphery structure. We started from a degree-corrected SBM, which naturally produces communities. However, a single-level SBM yields overly clean, well-separated clusters unlike those found in practice. To obtain more realistic community structures, we made the model multi-level. Finally, to capture core-periphery structure, we introduced low-degree peripheral nodes via a preferential attachment process. As our ablations (Table 5) show, using simpler and less realistic priors leads to notably worse performance. Following your comment, we will extend this discussion and would welcome suggestions.
>
> **W3:** Thank you for this suggestion. We agree that our study would benefit from an ablation of the MGM loss component. We have conducted this experiment, and present the results below. Please note that due to time and computational constraints, we provide results after the first stage of pretraining.
>
> ||artnet-exp|artnet-views|avazu-ctr|city-reviews|city-roads-M|hm-prices|tolokers-2|twitch-views|
> |:---|:---|:---|:---|:---|:---|:---|:---|:---|
> |GraphPFN (ICL)|50.61|60.05|28.31|79.46|61.65|73.37|59.04|64.23|
> |GraphPFN w/o MGM (ICL)|46.45|60.26|27.71|79.37|61.45|71.43|54.95|60.04|
>
> Removing MGM from pretraining notably degrades ICL performance on most datasets, with particularly large drops on `artnet-exp`, `tolokers-2`, and `twitch-views`.
>
> **W4:** Thank you for this suggestion. We conducted the requested experiments with GCOPE's prompt tuning (PT) variant. As shown in the table below, PT consistently underperforms full finetuning (FT) across all datasets except `pubmed`, where the difference is marginal. This is consistent with expectations: our evaluation uses non-few-shot splits (10%/10%/80% train/val/test), where the relatively large amount of labeled data favors full finetuning over lightweight adaptation. Therefore, the FT results reported in our paper already represent GCOPE's stronger configuration, and the comparison is fair.
>
> ||amazon-ratings|artnet-exp|city-reviews|facebook|pubmed|questions|tolokers-2|wiki-cs|
> |:---|:---|:---|:---|:---|:---|:---|:---|:---|
> |GCOPE (FT)|39.90 ± 0.43|14.92 ± 1.56|67.16 ± 0.98|85.08 ± 0.17|79.35 ± 0.70|6.59 ± 0.43|28.81 ± 1.28|59.13 ± 1.20|
> |GCOPE (PT)|37.48 ± 0.35|11.40 ± 0.60|55.36 ± 0.93|79.42 ± 0.84|79.93 ± 0.54|6.12 ± 0.71|25.77 ± 0.25|57.84 ± 1.92|

---

> > ### Author Rebuttal · Reviewer_1PoC · 2026-04-04
> >
> > I appreciate authors' response, which addresses my concerns on W3 and W4.
> >
> > W1 and W2 is related: the rebuttal still implies that the graph prior/synthetic graph generation is quite an adhoc process rather than systematic and principled. Given the diversity of graph data, the proposed prior may not generalize well across distinct graph types and/or domains.

---

> > > ### Author Response · Authors · 2026-04-07
> > >
> > > Thank you for your reply and for the continued engagement. We are glad that our response addressed your concerns on W3 and W4.
> > >
> > > Regarding prior design, it follows the general principles established in PFN literature [1, 2]: maximize diversity of the generated data while encoding useful inductive biases. Specifically, we sample diverse hyperparameters for each synthetic graph and encode structural inductive biases (complex community and core-periphery structure) motivated by well-known properties of real-world networks, as described in our rebuttal.
> > >
> > > We also believe the quality of a prior is best judged by its empirical performance across diverse settings. Within our scope, the evaluation covers a broad range of domains (social, road, co-reviewing, co-purchasing, citation networks, and others) and tasks (fraud detection, view count prediction, CTR prediction, price prediction, speed prediction, and more). GraphPFN shows strong results across all of them, suggesting that the prior generalizes well in practice.
> > >
> > > ***
> > >
> > > [1] Hollmann et al., "TabPFN: A transformer that solves small tabular classification problems in a second", ICLR (2023).
> > >
> > > [2] Qu et al., "TabICLv2: A better, faster, scalable, and open tabular foundation model", arXiv:2602.11139 (2026).

---

### Decision · Program_Chairs · 2026-04-30

**Decision:**

Accept (regular)

**Comment:**

The paper proposes GraphPFN, a prior-data fitted network (PFN) designed and pre-trained for graph node-level tasks. Building on the tabular foundation model LimiX, the authors augment each transformer block with attention-based graph message-passing adapters and design a graph prior combining multi-level stochastic block models with preferential-attachment structural causal models for attribute generation. It matches or outperforms both GCN-based foundation models (G2T-FM, TAG) and task-specific GNNs on most of the 13 evaluation datasets, and shows strong in-context learning performance without fine-tuning.

The paper is well-received as the first PFN designed natively for graph node-level tasks: reviewers highlight the principled combination of structural and attribute priors, the well-designed ablation study demonstrating that prior design is load-bearing, and strong in-context learning and fine-tuned results across diverse node-level graphs. The main reservation concerns the design of the graph prior and generalization to distinct graph types; the authors responded substantively, showing that simpler priors lead to notably worse performance, and added experiments on many-class settings, runtime, and scalability. The rebuttal meaningfully strengthened the submission, with most reviewers resolving or partially resolving their concerns.

For the camera-ready: explicit discussion of scalability limits and the mini-batch approximation; a clearer articulation of what the prior does and does not cover; a brief comparison to concurrent PFN-for-graphs efforts; and incorporation of the additional experimental results from the rebuttal into the main paper.